# Insights into the deterministic skill of air quality ensembles from the analysis of AQMEII data

Ioannis Kioutsioukis[ab], Ulas Im[c], Efisio Solazzo[b], Roberto Bianconi[d], Alba Badia[e], Alessandra Balzarini[f], Rocío Baró[l], Roberto Bellasio[d], Dominik Brunner[g], Charles Chemel[h], Gabriele Curci[ij], Hugo Denier van der Gon[k], Johannes Flemming[m], Renate Forkel[n], Lea Giordano[g], Pedro Jiménez-Guerrero[l], Marcus Hirtl[o], Oriol Jorba[e], Astrid Manders-Groot[k], Lucy Neal[p], Juan L. Pérez[q], Guidio Pirovano[f], Roberto San Jose[q], Nicholas Savage[p], Wolfram Schroder[r], Ranjeet S Sokhi[h], Dimiter Syrakov[s], Paolo Tuccella[ij], Johannes Werhahn[n], Ralf Wolke[r], Christian Hogrefe[t], Stefano Galmarini[b]

a.  University of Patras, Department of Physics, University Campus 26504 Rio, Greece.
b.  European Commission, Joint Research Centre, Directorate for Energy, Transport and Climate, Air and Climate Unit, Ispra (VA), Italy.
c.  Aarhus University, Department of Environmental Science, Roskilde, Denmark
d.  Enviroware srl, Concorezzo (MB), Italy.
e.  Earth Sciences Department, Barcelona Supercomputing Center (BSC-CNS), Barcelona, Spain.
f.  Ricerca sul Sistema Energetico (RSE) SpA, Milan, Italy
g.  Laboratory for Air Pollution and Environmental Technology, Empa, Dubendorf, Switzerland.
h.  Centre for Atmospheric & Instrumentation Research, University of Hertfordshire, College Lane, Hatfield, AL10 9AB, UK.
i.  Department of Physical and Chemical Sciences, University of L'Aquila, L'Aquila, Italy.
j.  Center of Excellence for the forecast of Severe Weather (CETEMPS), University of L'Aquila, L'Aquila, Italy.
k.  Netherlands Organization for Applied Scientific Research (TNO), Utrecht, The Netherlands.
l.  University of Murcia, Department of Physics, Physics of the Earth. Campus de Espinardo, Ed. CIOyN, 30100 Murcia, Spain.
m.  ECMWF, Shinfield Park, RG2 9AX Reading, United Kingdom.
n.  Karlsruher Institut für Technologie (KIT), IMK-IFU, Kreuzeckbahnstr. 19, 82467 Garmisch-Partenkirchen, Germany.
o.  Zentralanstalt für Meteorologie und Geodynamik, ZAMG, 1190 Wien, Austria.
p.  Met Office, FitzRoy Road, Exeter, EX1 3PB, United Kingdom.
q.  Environmental Software and Modelling Group, Computer Science School - Technical University of Madrid, Campus de Montegancedo - Boadilla del Monte-28660, Madrid, Spain.
r.  Leibniz Institute for Tropospheric Research, Permoserstr. 15, D-04318 Leipzig, Germany.
s.  National Institute of Meteorology and Hydrology, Bulgarian Academy of Sciences, 66 Tzarigradsko shaussee Blvd., Sofia 1784, Bulgaria.
t.  Atmospheric Modelling and Analysis Division, Environmental Protection Agency, Research Triangle Park, USA.

**Abstract**
Simulations from chemical weather models are subject to uncertainties in the input data (e.g.
emission inventory, initial and boundary conditions) as well as those intrinsic to the model
(e.g. physical parameterization, chemical mechanism). Multi-model ensembles can improve
the forecast skill provided that certain mathematical conditions are fulfilled. In this work, four
ensemble methods were applied to two different datasets and their performance was compared
for ozone ($O_3$), nitrogen dioxide ($NO_2$) and particulate matter ($PM_{10}$). Apart from the
unconditional ensemble average, the approach behind the other three methods relies on
adding optimum weights to members or constraining the ensemble to those members that
meet certain conditions in time or frequency domain. The two different datasets were created
for the first and second phase of the Air Quality Model Evaluation International Initiative
(AQMEII). The methods are evaluated against ground level observations collected from the
EMEP and Airbase databases. The goal of the study is to quantify to what extent we can
extract predictable signals from an ensemble with superior skill over the single models and
the ensemble mean. Verification statistics shows that the deterministic models simulate better
$O_3$ than $NO_2$ and $PM_{10}$, linked to different levels of complexity in the represented processes.
The unconditional ensemble mean achieves higher skill compared to each station's best
deterministic model at no more than 60% of the sites, indicating for the rest a combination of
members with unbalanced skill difference and error dependence. The promotion of the right
amount of accuracy and diversity within the ensemble results in an average additional skill up
to 31% compared to using the full ensemble in an unconditional way. The skill improvements
were higher for $O_3$ and lower for $PM_{10}$, associated to the extent of potential changes in the
joint distribution of accuracy and diversity in the respective ensembles. The skill
enhancement was superior using the weighting scheme but the training period required to
acquire representative weights was longer compared to the sub-selecting schemes. Further
development of the method is discussed in the conclusion.
Keywords: AQMEII, multi-model ensembles, air quality model, error decomposition,
verification.

# 1 Introduction

Uncertainties in atmospheric models such as the chemical weather models, whether due to the input data or the model itself, limit the predictive skill. The incorporation of data assimilation techniques and the continued effort in understanding the physical, chemical and dynamical processes, result in better forecasts (Zhang et al., 2012). In addition, ensemble methods provide an extra channel for forecast improvement and uncertainty quantification. The benefits from ensemble averaging arise from filtering out the components of the forecast with uncorrelated errors (Kalnay, 2003).

The European Centre for Medium-Range Weather Forecast (ECMWF) reports an increase in forecast skill of 1 day per decade for meteorological variables, evaluated on the geopotential height anomaly (Simmons, 2011). The air quality modelling and monitoring has a shorter history that does not allow a similar adequate estimation of such trend for the numerous species being modelled. Moreover, the skill changes dramatically from species to species strongly connected to the availability of accurate emission data. Results for ozone suggest that medium-range forecasts can be performed with a quality similar to the geopotential height anomaly forecasts (Eskes et al., 2002). Besides the continuous increase in skill due to the improved scientific understanding, harmonized emission inventories, more accurate and denser observations as well as ensemble averaging, an extra gain of similar magnitude can be achieved for ensemble-based deterministic modelling using conditional averaging (e.g., Galmarini et al., 2013; Mallet et al., 2009; Solazzo et al., 2013).

Ideally, for continuous and unbiased variables, the multi-model ensemble mean outscores the skill of the deterministic models provided that the members have similar skill and independent errors (Potempski and Galmarini, 2009; Weigel et al., 2010). Practically, the multi-model ensemble mean usually outscores the skill of the deterministic models if the evaluation is performed over multiple observation sites and times. This occurs because over a network of stations, there are some where the essential conditions (e.g. the skill difference between the models is not too large) for the ensemble members are fulfilled, favouring the ensemble mean; for the remaining stations, where the conditions are not fulfilled, local verification identifies the best model but generally no single model is the best at all sites. Hence, although the skill of the numerical models varies in space (latitude, longitude, altitude) and time (e.g., hour of the day, month, season), the ensemble mean is usually the most accurate spatio-temporal representation.

One of the challenges in multi-model ensemble forecasting is the processing of the deterministic models datasets prior to averaging in order to construct another dataset for which its members ideally constitute an *independent and identically distributed* (i.i.d.) sample (Kioutsioukis and Galmarini, 2014; Bishop and Abramowitz, 2013). This statistical process favours the ensemble mean at each observation site. Two basic pathways exist to achieve this goal: model weighting or model sub-selecting. There are several methods to assign weights to ensemble members such as the singular value decomposition (Pagowski et al., 2005), dynamic linear regression (Pagowski et al., 2006; Djalalova et al., 2010), Kalman filtering (Delle Monache et al., 2011), Bayesian model averaging (Riccio et al., 2007; Monteiro et al., 2013) and analytical optimization (Potempski and Galmarini, 2009) while model selection usually relies on the quadratic error or its proxies, in time (e.g. Solazzo et al., 2013; Kioutsioukis and Galmarini., 2014) or frequency space (Galmarini et al., 2013). The majority of those ensemble studies focuses on $O_3$ and only recently the studies also involve particulate matter (Djalalova et al., 2010; Monteiro et al., 2013).

In this work, we apply and intercompare both approaches (weighting and sub-selecting) using the Air Quality Model Evaluation International Initiative (AQMEII) datasets from phase I and phase II. The ensemble approaches are evaluated against ground level observations from the EMEP and Airbase databases, focusing on the pollutants $O_3$, $NO_2$ and $PM_{10}$ that exhibit different levels of forecast skill. The differences between the multi-model ensembles of phase I (hereafter AQMEII-I) and phase II (hereafter AQMEII-II) originate from many sources, related to both the input data and the models: (a) the simulated years are different (2006 vs. 2010), therefore the meteorological conditions are different; (b) emission methodologies have changed; (c) boundary conditions are very different; (d) the composition of the ensembles is different; (e) the models in AQMEII-II use on-line coupling between meteorology and chemistry; (f) the models may have been updated with new science processes apart from feedback processes. The uncertainties arising from observational errors are not taken into consideration.

In spite of these differences we consider the analysis of the two sets of ensembles revealing. In detail, the objectives of the paper are (a) to interpret the skill of the unconditional multi-model mean within AQMEII-I and AQMEII-II (b) to calculate the maximum expectations in the skill of alternative ensemble estimators and (c) to evaluate the operational implementation of the approaches using cross-validation. The originality of the study includes: (a) the

comparison of several ensemble methods on pollutants of different skill using different
datasets, (b) the introduction of an approach based on high-dimension spectral optimization,
(c) the introduction of innovative charts for the interpretation of the error of the unconditional
ensemble mean with respect to indicators reflecting the skill difference and error dependence
of the models as well as the effective number of models. Therefore we carry out an analysis of
the performance of different ensemble techniques rather than a comparison of the results from
the two phases of the AQMEII activity.
The paper is structured as follows: section 2 provides a brief description of the ensemble's
basic properties through a series of conditions expressed by mathematical equations. In
section 3, the experimental setup is described. Results are presented in section 4, where the
skill of the deterministic models, the unconditional ensemble mean and the conditional
ensemble estimators are analysed and intercompared. Conclusions are drawn in Section 5.

## 2   Minimization of the ensemble error

The notation conventions used in this section are briefly presented in the following. Assuming
an ensemble composed of M members (i.e. output of modelling systems) denoted as $f_i$,
$i=1,2,...,M$, the multi-model ensemble mean can be evaluated from $\bar{f} = \sum_{i=1}^{M} w_i f_i$, $\sum w_i = 1$. The
weights ($w_i$) sum up to one and can be either equal (uniform ensemble) or unequal
(nonuniform ensemble). The desired value (measurement) is $\mu$.
Assuming a uniform ensemble, the squared error (MSE) of the multi-model ensemble mean
can be broken down into three components, namely, the average bias (1st term), the average
error variance (2nd term) and the average error covariance (3rd term) of the ensemble members
(Ueda and Nakano, 1996):

$$MSE(\bar{f}) = \left(\frac{1}{M}\sum_{i=1}^{M}(f_i - \mu)\right)^2 + \frac{1}{M}\left(\frac{1}{M}\sum_{i=1}^{M}(f_i - \mu)^2\right)$$
$$+ \left(1 - \frac{1}{M}\right)\left(\frac{1}{M(M-1)}\sum_{i=1}^{M}\sum_{i \neq j}(f_i - \mu)(f_j - \mu)\right)$$

      Eq.1

The decomposition provides the reasoning behind ensemble averaging: as we include more
ensemble members, the variance factor is monotonically decreasing and the MSE converges
towards the covariance factor. Covariance, unlike the other two positive definite terms, can be
either positive or negative; its minimization requires an ensemble composed by independent
or even better, negatively correlated members. In addition, bias correction should be a
necessary step prior to any ensemble manipulation. More details regarding this decomposition
within the air quality ensembles context can be found in Kioutsioukis and Galmarini, 2014.
In a similar fashion, the squared error of the multi-model ensemble mean can be decomposed
into the difference of two positive-definite components, with their expectations characterized
as accuracy and diversity (Krogh and Vedelsby, 1995):

$$\boldsymbol{MSE}(\bar{\boldsymbol{f}}) = \frac{1}{M}\sum_{i=1}^{M}(f_i - \mu)^2 - \frac{1}{M}\sum_{i=1}^{M}(f_i - \bar{f})^2 \qquad \textbf{Eq.2}$$

This decomposition proves that the error of the ensemble mean is guaranteed to be less than
or equal to the average quadratic error of the component models. The minimum ensemble
error depends on the right trade-off between accuracy (1st term on the r.h.s. of Eq. 2) and
diversity (2nd term on the r.h.s. of Eq. 2). If the evaluation is applied on multiple sites, then the
equations 1 and 2 should be replaced with their expectations over the stations.
An error decomposition approach can also be applied on the spectral components (SC) of the
observed and modelled time-series. The data can be spectrally decomposed with the
Kolmogorov-Zurbenko (*kz*) filter (Zurbenko, 1986) while the original time-series can be
obtained with the linear combination of the spectral components. Assuming the pollution data
at the frequency domain yields N principal spectral bands, the squared error of the multi-
model ensemble mean can be broken down into $N^2$ components (Galmarini et al., 2013;
Solazzo and Galmarini, 2016):

$$\boldsymbol{MSE}(\bar{\boldsymbol{f}}) = \sum_{i=1}^{N} MSE\left(SC_{\bar{f}_i}\right) + \sum_{i \neq j} Cov\left(SC_{\bar{f}_i}, SC_{\bar{f}_j}\right) \qquad \textbf{Eq.3}$$

This decomposition shows that the error of the ensemble mean could be split into the sum of
N errors associated with different parts of the spectrum (1st term), provided the spectral
components are independent (the covariance term is zero). The minimization of the error at
each spectral band can be achieved with another approach such as the decompositions
presented in Eq.1 and Eq.2.
The three decompositions presented assume uniform ensembles, i.e. all members receive
equal weight. For the case of a non-uniform ensemble, the MSE of the multi-model ensemble
mean can be analytically minimized to yield the optimal weights, provided that the
participating models are bias-corrected (Potempski and Galmarini, 2009):

$$\overline{w} = \frac{K^{-1}l}{(K^{-1}l, l)}$$   **Eq.4**

where, $w$ is the vector of optimal weights, $K$ is the error covariance matrix and $l$ the unitary
vector. In its simplest form, the equation assigns one weight for each model at each
measurement site; more complicated versions like multidimensional optimisation for many
variables (e.g. chemical compounds) at many sites simultaneously are not discussed here.
Unlike the straightforward calculation of the optimal weights, the sub-selecting schemes make
use of a reduced-dimensionality ensemble. An estimate of the effective number of models
($N_{EFF}$) sufficient to reproduce the variability of the full ensemble is calculated as (Bretherton
et al., 1999):

$$N_{EFF} = \frac{\left(\sum_{i=1}^{M} s_i\right)^2}{\sum_{i=1}^{M} s_i^2}$$   **Eq.5**

where $s_i$ is eigenvalue of the error covariance matrix. Theoretical evidence shows that the
fraction of the overall variance expressed by the first $N_{EFF}$ eigenvalues is 86%, provided that
the modelled and observed fields are normally distributed (Bretherton et al., 1999). The
highest eigenvalue is denoted as $s_m$.
It is apparent from the above considerations that the skill of the unconditional ensemble mean
has the potential for certain advantages over the single members, provided some properties
are satisfied. As those properties are not systematically met in practice, superior ensemble
skill can be achieved through sub-selecting or weighting schemes presented in this section.
An inter-comparison of the following approaches in ensemble averaging is investigated in this
work using observed and simulated air quality time-series:
• Unconditional ensemble mean *(mme)*
• Conditional (on selected members) ensemble mean in time domain *(mme<)*: the
optimal trade-off between accuracy and diversity (equation 2) is identified across all
possible combinations of the available M models (Kioutsioukis and Galmarini, 2014).

The number of members in the ensemble combination that gives the minimum error will be used as the effective number of models ($N_{EFF}$) rather than its estimate based on the independent components of the ensemble (eq. 5).

- Conditional (on selected members) ensemble mean in frequency domain *(kzFO)*: following equation 3, an ensemble estimator is synthesized from the best member at each spectral band (Galmarini et al., 2013). The original time-series are decomposed into four spectral components (see Appendix I), namely the intra-diurnal, diurnal, synoptic and long-term component, using the Kolmogorov-Zurbenko filter (Zurbenko, 1986).

- Conditional (on selected members) ensemble mean in frequency domain *(kzHO)*: it is an extension of the *kzFO,* where the spectral components of the ensemble estimator are averaged from $N_{EFF}$ members at each spectral band (rather than the best).

- Conditional (optimally weighted) ensemble mean *(mmW):* according to equation 4 (Potempski and Galmarini, 2009).

The skill of the models and the examined ensemble averages have been scored with the following statistical parameters: (1) normalised mean square error (NMSE), i.e. the mean square error (MSE) divided by $\bar{O}\,\bar{M}$, where $\bar{O}$ and $\bar{M}$ are the mean value of the observation and the model respectively, (2) probability of detection (POD) and false alarm rate (FAR), i.e. the proportion of occurrences (e.g. events exceeding threshold value) that were correctly identified and the proportion of non-occurrences that were incorrectly identified respectively (3) Taylor plots (Taylor, 2001), which summarize standard deviation, root mean square error (RMSE) and Pearson product-moment correlation coefficient in a single point on a two-dimensional plot.

## 3    Setup: experiments, models and observations

The two AQMEII ensemble datasets have simulated the air quality for Europe [(-10,39)W; (30,65)N] and North America [(-125,-55)W; (26,51)N]. Despite the common domains, the modelling systems across the two phases have profound differences. The simulation year was 2006 for AQMEII-I and 2010 for AQMEII-II, therefore the two sets are dissimilar with respect to the input data (emissions, chemical boundary conditions, meteorology). Boundary conditions are obtained from GEMS (Global and Regional Earth-System Monitoring using Satellite and in-situ data) in AQMEII-I and MACC (Monitoring Atmospheric Composition &

Climate) in AQMEII-II. The air quality models of the second phase are coupled with their
meteorological driver (chemistry feedbacks on meteorology), while those of the first phase
are not. The participating models are also different. Detailed analysis of the emissions,
boundary conditions and meteorology for the modelled year 2006 (AQMEII-I) is presented in
Pouliot et al. (2012), Schere et al. (2012) and Vautard et al. (2012). For 2010 (AQMEII-II),
similar information is presented in Pouliot et al. (2015), Giordano et al. (2015) and Brunner et
al. (2015).
The participating models follow a restrictive protocol concerning the emissions and the
meteorological and chemical boundary conditions. In AQMEII-I, meteorological models
applied nudging to the NCEP GFS meteorological analysis. In AQMEII-II, the simulations
were run more in a way as if they were real forecasts; meteorological boundary conditions for
the majority of the models were from the ECMWF operational archive (see Tables 1 and 2 in
Brunner et al, 2015) and no nudging or FDDA was applied. However, the driving
meteorological data were analysis (but no reanalysis) for all simulations, with exception of the
COSMO-MUSCAT run. Hence, the runs from AQMEII-II are more like forecasts than those
from AQMEII-I.
Recent studies with regional air quality models yielded that the full variability of the
ensemble can be retained with only an effective number of models ($N_{EFF}$) on the order of 5-6
(e.g. Solazzo et al., 2013; Kioutsioukis and Galmarini, 2014; Marecal et al., 2015). The
minimum number of ensemble members to sample the uncertainty should be well above $N_{EFF}$;
for this reason, we focus on the European domain (EU) due to its sufficient number of models
to form the ensemble.
Table 1 summarises the features of the modelling systems analysed in this study with regard to
$O_3$, $NO_2$ and $PM_{10}$ concentrations in the EU. The modelling contribution to the two phases of
AQMEII consists of 12, 13 and 10 models for $O_3$, $NO_2$ and $PM_{10}$ respectively in AQMEII-I,
while 14 members were available for all species in AQMEII-II. Several discrete simulations
of WRF-Chem with alternative chemistry and physics configurations are included in
AQMEII-II (Forkel et al. 2015, San José et al, 2015, Baró et al., 2015).
Following the statements of section 2, each model has been bias-corrected prior to the
analysis, i.e. its own mean bias over the examined three-month period has been subtracted
from its modelled time-series at each monitoring site. For each modelling system, its long-
term systematic error is a known quantity estimated during its validation stage; therefore the

subtraction of the seasonal bias does not restrict the generality of the study. Actually, the requirement for bias removal is a necessary condition only for the weighted ensemble mean. In the results section we will address this issue and its effect on the skill of the ensemble estimators.

The observational data sets for $O_3$, $NO_2$ and $PM_{10}$ derived from the surface AQ monitoring networks operating in the EU constitutes the same data set used in the first and second phases of AQMEII to support model evaluation. All monitoring stations are rural and have data at least 75% of the time. The network is denser for $O_3$ (451/450 stations in AQMEII-I/II) for which there are as many monitoring stations as for $NO_2$ (290/337 stations in AQMEII-I/II) and $PM_{10}$ (126/131 stations in AQMEII-I/II) combined, with $PM_{10}$ having the fewest observations. **Figure 1** compares the statistical distribution of all three species between the two AQMEII phases, through the cumulative density function composed from the mean value at each percentile of the observations. The Kolmogorov-Smirnov test (Massey, 1951) yields that only the $PM_{10}$ distributions differ at the 1% significance level. It results from the unavailability of data for France and UK in AQMEII-II for $PM_{10}$ (station locations are shown in **Figure 3**).

## 4    Results

In this section we apply the conceptual context briefly presented in section 2 to investigate the effect of the differences in the ensemble properties within each of the two AQMEII phases (Rao et al., 2011) in the skill of the unconditional multi-model mean. The potential for improved estimates through conditional ensemble averages and their robustness is ultimately assessed.

From the provided station-based hourly time-series, we analysed one season (three-monthly period) with continuous data and relatively high concentrations; for $O_3$, June-July-August was selected while September-October-November is used for $NO_2$ and $PM_{10}$.

### 4.1    Single Models

The distributions of each model's NMSE for $O_3$, $NO_2$ and $PM_{10}$ over all monitoring stations are presented in Figure 2 as box-and-whisker plots. On each box, the central mark indicates the median, and the bottom and top edges of the box indicate the 25th and 75th percentiles,

respectively. The whiskers extend to the most extreme data points not considered outliers (i.e. points with distance from the $25^{th}$ and $75^{th}$ percentiles smaller than 1.5 times the interquartile range). Among the examined pollutants, the models simulate better the $O_3$ concentrations, as is evident from the axis scale. The highest variability in the skill between and within the models is observed for $NO_2$.

The distribution of average NMSE at each station (<NMSE>) has a median on the order of 0.1 for $O_3$ and 0.5 for $NO_2$ and $PM_{10}$ for both phases (Table 2). The application of the Kolmogorov-Smirnov test (Massey, 1951) on the <NMSE> distributions across AQMEII-I and AQMEII-II shows that there are no statistically significant differences in the <NMSE> distributions between the two ensemble datasets at the 1% significance level. The same also applies for the statistical distribution of the minimum NMSE at each station (NMSE$_{BEST}$) at each monitoring station. Hence, despite the different modelling systems and input data, the <NMSE> and NMSE$_{BEST}$ distributions between AQMEII-I and AQMEII-II are indistinguishable for the three examined pollutants.

Besides <NMSE> and NMSE$_{BEST}$, we evaluate the percentage of cases each model has been identified as being 'best' and calculate the coefficient of variation (*CoV=std/mean*) of this index for each ensemble. If models were behaving like *i.i.d.*, the probabilities of being best would be roughly equal (~1/M) for all models and the *CoV* would generally be well below unity for the examined range of ensemble members. As can be inferred from Table 2, the proportion of *equally good models* is higher for $O_3$ and $NO_2$ in the $2^{nd}$ dataset. Among the pollutants, the *CoV* of $NO_2$ exhibits the most dramatic change.

## 4.2   Pitfalls of the unconditional multi-model mean

The skill of the multi-model mean has been compared against the skill of the best deterministic model, independently evaluated at each monitoring site (hereafter *bestL*). The geographical distribution of the ratio RMSE(*mme*)/RMSE$_{BESTMODEL}$ is presented in Figure 3. The indicator does not exhibit any longitudinal or latitudinal dependence. Summary statistics indicate that the *mme* outscores the *bestL* at roughly half of the stations for $O_3$ (namely 52/49 for AQMEII-I/II) and at approximately 40% of the stations for $PM_{10}$ (38/42). The same statistic for $NO_2$ varies considerably (39/64). The Kolmogorov-Smirnov test yields that the corresponding distributions (pI/pII) are different at the 1% significance level but the t-test demonstrates that the mean of the distributions differ significantly only for $NO_2$. The reason

behind the skill of *mme* with respect to the *bestL* is investigated next with respect to the skill
difference and the error dependence of each ensemble.
The skill difference between the best model and the average skill is inferred from the
indicator $NMSE_{BEST}$ /<NMSE> (Table 2). High values of the indicator correspond to small
skill differences between the ensemble members (desirable). The distribution of the
$NMSE_{BEST}$ /<NMSE> at each station has a median on the order of 0.6-0.8, variable with
respect to the dataset and the pollutant. The spread of the indicator, measured by its
interquartile range, is higher for $NO_2$ and lower for $O_3$.
The eigenvalues of the covariance matrix calculated from the model errors provides
information on the members' diversity and the ensemble redundancy (Eq. 5). Following the
eigen-analysis of the error covariance matrix at each station separately and converting the
eigenvalues to cumulative amount of explained variance, the resulting matrix is presented into
box and whisker plot (Figure 4). The error dependence of the ensemble members is deduced
from the explained variation by the maximum eigenvalue $s_m$. Low values of the indicator
corresponds to independent members with small error dependence (desirable). The average
variation explained by $s_m$ ranges between 65% and 79%, taking the lower values for $NO_2$. The
spread of the indicator, measured by its interquartile range, is higher for $NO_2$ and lower for
$O_3$.
All species demonstrate smaller skill difference and higher error dependence in the AQMEII-
II dataset. The Kolmogorov-Smirnov test yielded the difference in the corresponding
distributions of the indicators between AQMEII-I and AQMEII-II is significant at the 1%
level. However, it is the joint distribution of skill difference and error dependence that
modulates the *mme* skill with respect to the *bestL,* as seen in Figure 5. Shifts in the
distributions of the indicators at opposite directions eventually cancel out, yielding no change
in the *mme* skill. This case is observed for $O_3$ and $PM_{10}$. For $NO_2$, skill difference was
improved more than error dependence was worsened, yielding a net improvement of *mme* in
AQMEII-II.
The area below the diagonal in Figure 5 corresponds to monitoring sites with disproportionally
low diversity under the current level of accuracy. This area of the chart indicates high spread
in skill difference and relatively highly dependent errors. This situation practically means a
limited number of skilled models with correlated errors, which in turn denotes a small $N_{EFF}$
value as demonstrated in Figure 6. The opposite state is true for the area above the diagonal. It
corresponds to locations that are constituted from models with comparable skill and relatively
independent errors, reflecting a high $N_{EFF}$ value. This matches the desired synthesis for an
ensemble.
The cumulative distribution of $N_{EFF}$ from the error minimization (i.e. the optimal trade-off
between accuracy and diversity) across all possible combinations of M models at each site is
also presented in Figure 4 (solid line). At over 90% of the stations, we do not need more than 5
members for $O_3$, 6 members for $PM_{10}$ and 6-7 members for $NO_2$. Further, from a pool of 10-
14 models, the benefits of ensemble averaging cease after 5-7 members (but not 5-7 particular
members across all stations).

## 4.3 Conditional multi-model mean

Following the identification of the weaknesses in the ensemble design, the potential for
corrections through more sophisticated schemes is now investigated. We consider the skill of
the multi model mean as the starting point and we investigate pathways for further enhancing
it through the non-trivial problem of weighting or sub-selecting. The optimal weights (*mmW*)
are estimated from the analytical formulas presented in Potempski and Galmarini, 2009. The
sub-selection of members has been built upon the optimization of either the accuracy/diversity
trade-off (*mme<*) (Kioutsioukis and Galmarini, 2014) or the spectral representation of 1[st]
order components by different models (*kzFO*) (Galmarini et al., 2013). Another approach
built upon higher order (namely, $N_{EFF}$) spectral components (*kzHO*) is also investigated. In
this section we mark the boundaries of the possible improvements for different ensemble
mean estimators applicable to the AQMEII datasets and their sensitivity to sub-optimal
conditions using cross-validation.
The global skill of all the single models and the ensemble estimators, evaluated at all stations,
are presented in Figure 7 in the form of Taylor plots. For $O_3$, the deterministic models have
standard deviations that are smaller compared to observations and a narrow correlation pattern
(~0.7) that is slightly deteriorated in AQMEII-II. For $NO_2$, members with higher variance -as
well as lower- than the observed variance exist in the ensemble while the correlation spread is
becoming narrower in AQMEII-II and demonstrates a minor improvement. Last, simulated
$PM_{10}$ from the deterministic models displays smaller standard deviation compared to
observations with a wide correlation spread (0.3-0.6). The multi-model mean is always found
closer to the reference point, in an area that incorporates lower error and increased correlation
but at the same time generally low variance. The examined ensemble estimators (*mmW,*
*mme<, kzFO, kzHO*) are horizontally shifted from *mme*, hence they demonstrate even lower
error and increased correlation and variance. Among them, the highest composite skill was
found for *mmW*, followed by *kzHO*.
A comparison between the skill of the examined ensemble estimators versus the *mme* and the
best single model is now conducted (Table 3). The best single model is evaluated globally
(*bestG*: average across all stations) and locally (*bestL*: at each station separately). The former
estimates the best average deterministic skill among the candidate models; the latter provides
a useful indicator for controlling whether the anticipated benefits of ensemble averaging
holds. The skill scores have been evaluated against the guaranteed minimum gain of the
ensemble (<MSE>), the ensemble mean (*mme*) and the best single model globally (*bestG*).
The estimations calculated from the unprecedented AQMEII datasets (2 years of hourly
measurements and simulations from 2 different ensembles of 10-14 models each at over 450
stations for 3 pollutants) allows the following interpretation:
- The *mme* always achieves lower error than *bestG*. The advancement is higher for $O_3$
(9-22%), followed by $NO_2$ (7-9%) while the $PM_{10}$ demonstrate the least skill
improvement (1-3%). With respect to *bestL*, the *mme* generally attains similar or
slightly higher MSE. Hence, the average error over multiple stations statistically
favours the ensemble mean over the single models but the comparison at each site
generally does not as it depends on the skill difference and the error dependence of the
models.
- The skill score of *mme* over <MSE> (i.e., the guaranteed upper ceiling for the MSE of
*mme,* from eq. 2) ranges between 15% and 30%, higher for $NO_2$ and lower for $PM_{10}$.
According to eq. 2, this number also represents the diversity as percentage of the
accuracy. Therefore, besides improving the single models, their combination in an
ensemble confines the *mme* skill if their diversity is limited.
- The skill score of the examined ensemble estimators (*mmW, mme<, kzFO, kzHO*) over
<MSE> ranges between 25% and 50%, higher for $O_3$ and $NO_2$ and lower for $PM_{10}$.
Among them, the improvement is higher for *mmW* and lower for *mme<* and *kzFO*.
Thus, the promotion of accuracy and diversity within the ensemble almost doubles the
distance to <MSE> compared to *mme* and results in an additional skill over the *mme*
between 14% and 31% (for *mmW*).

1      - The improvement of the ensemble estimator using selected $N_{EFF}$ members (*mme<*)
2         over all members (*mme*) is illustrated in Figure 8 in the context of skill difference and
3         error dependence. The charts demonstrate no points below the diagonal, i.e. the sub-
4         selection results in an ensemble constituted from models with comparable skill and
5         relatively independent errors (compared to the full ensemble).
- The theoretical minimum MSE of *mme* for the case of unbiased and uncorrelated
7         models (from eq. 1) is far from being achieved from all ensemble estimators.

The statistical distributions of the skill scores of the examined ensemble estimators (*mmW,*
*mme<, kzFO, kzHO*) over *mme* are well bounded from above to lower than unity values
(Figure 9). The only exception exists for roughly 10% of the stations, for all pollutants, where
*kzFO* demonstrates higher MSE compared to *mme*. Unlike the other ensemble estimators,
*kzFO* utilises independent spectral components each obtained from a single model,
eliminating the possibility for 'cancelling out' of random errors. All cases belonging to this
10% of the samples (lower tail of the cdf) demonstrate high $N_{EFF}$, where the benefits from
unconditional ensemble averaging are optimal (Kioutsioukis and Galmarini, 2014). Contrary,
for another 10% of the stations (upper tail of the cdf), there is an abrupt improvement from
the conditional ensemble estimators. Those cases demonstrate low $N_{EFF}$, where the benefits
from unconditional ensemble averaging are minimal.
The ability to simulate extreme values is now examined through the POD and FAR indices.
Two thresholds were utilised for each pollutant, being 120 and 180 μg/m$^3$ for $O_3$, 25 and 50
μg/m$^3$ for $NO_2$ and 50 and 90 μg/m$^3$ for $PM_{10}$. The average 90$^{th}$ percentile across the stations
was 129/117 μg/m$^3$ (AQMEII-I/II) for $O_3$, 30/26 μg/m$^3$ for $NO_2$ and 52/33 μg/m$^3$ for $PM_{10}$
(Figure 1). Hence, the thresholds fall into the upper 10% of the distributions, being even more
extreme for $PM_{10}$ in AQMEII-II. The numbers in Table 4 give rise to the following inferences:
- for $O_3$ and $NO_2$, *mme* achieves somewhat higher POD than *bestG* at the lower
threshold but the order is reversed at the higher threshold. For $PM_{10}$, *bestG* always
performs better than *mme* for values exceeding the lower threshold. As we move
towards the tail, the POD of *bestG* dominates over the *mme*. Thus, the ranking of the
*mme* and *bestG* at the extreme percentiles and on average (seen earlier) are opposite.
- The *mme<* generally achieves somewhat higher POD than *bestL* at the lower threshold
but the order is reversed at the higher threshold. Over that level, *kzFO* and *mmW* are
the only estimators with POD higher than *bestL*.

- As we move towards higher percentiles, the 1$^{st}$ order spectral model (*kzFO*) has higher POD than the higher-order spectral model (*kzHO*) due to the averaging in the latter. In addition, the frequency domain averaging (*kzHO*) had slightly higher POD compared to the time domain averaging (*mme<*).
- The *mmW*, besides its lower MSE, has the highest POD among all models and ensemble estimators.
- The variation of FAR was very small between all examined models and ensemble estimators.

The combination of the results from the average error and the extremes identifies *mmW* as the estimator that outscores the others across all percentiles. *kzFO* has high capacity for extremes but requires attention for the limited sites with high $N_{EFF}$, where its skill is inferior to *mme*. *kzHO* and *mme<* have both high skill across all percentiles (better for *kzHO*) but they could have inferior POD compared to *bestL* at the extreme percentiles. With respect to the pollutants, the advancement of *mmW* skill over *mme* was higher for $O_3$.

The additional skill over *mme* in the range between 8% and 31% from the statistical approaches applied to a pool of ensemble simulations identifies the upper ceiling of the improvements from the corrections in the skill difference and the error dependence of the ensemble members. The bound results from the removal of the seasonal bias from the time series and the optimal training of the methods. We now proceed with splitting the datasets into training and testing and explore the sensitivity of the *mmW* skill arising from improper bias removal and weights. Both factors are estimated on the training set for variable time-series length that is progressively increasing from 1 to 60 days, for all monitoring stations and pollutants. The evaluation period for all training windows is the same 30-day segment, not available in the training procedure. The analysis will provide a perspective on applying the techniques in a forecasting context, although the examined simulations did not operate in forecasting mode.

The interquartile range of the day-to-day difference in the weights is calculated and its range over all stations is displayed in Figure 10. No convergence occurs, however the variability of the *mmW* weights is notably reduced after a certain amount of time. If we set a tolerance level at the second decimal, to be satisfied at all stations, we need at a minimum 20-45 days of hourly time-series. The variability of weights is smaller for $O_3$ and higher for $NO_2$ and $PM_{10}$, explained by the larger NMSE spread in the latter case. The identification of the necessary

training or learning period will be assessed by its effect on the *mmW* skill. Table 5 presents the
*mmW* skill obtained from training over time series of different lengths varying from 5 to 60
days. For $O_3$, *mmW* trained over 10 days yields similar results with *mme* while longer periods
result in large departures from *mme*. $NO_2$ and $PM_{10}$ require larger training periods than $O_3$.
The use of *mmW* is practically of no benefit compared to *mme* if the traning period is less than
20 days for $NO_2$ and 30 days for $PM_{10}$. For all pollutants, the variability of the weights and
the bias has no effect in the error after 60 days.
The results demonstrate that the ensemble estimators based on the analytical optimization
become insensitive to inaccuracies in the bias and weights for training periods exceeding 60
days. Other published studies with weighted ensembles using non-analytical optimization
though (e.g. linear regression, Monteiro et al., 2012), argue that one month is sufficient for the
weights and the bias. The sub-selecting schemes are more robust compared to the optimal
weighting scheme in the variations of their parameters (bias, members). Using data from
AQMEII-I, training periods in the order of a week were found essential for *mme<*
(Kioutsioukis and Galmarini, 2014) and *kzFO* (Galmarini et al., 2013). Therefore, the
operational implementation of each ensemble approach requires knowledge of its safety
margins for the examined pollutants.
**5   Conclusions**
In this paper we analyze two independent suites of chemical weather modelling systems
regarding their effect in the skill of the ensemble mean (*mme*). The results are interpreted with
respect to the error decomposition of the *mme*. Four ways to extract more information from an
ensemble besides the *mme* are ultimately investigated and evaluated. The first approach
applies optimal weights to the models of the ensemble (*mmW*) and the other three methods
utilise selected members in time (*mme<*) or frequency (*kzFO, kzHO*) domain. The study
focuses on $O_3$, $NO_2$ and $PM_{10}$, using the unprecedented datasets from two phases of AQMEII
over the European domain.
The comparison of the *mme* skill versus the globally best single model (*bestG:* identified from
the evaluation over all stations)*,* points out that *mme* achieves lower average (across all
stations) error compared to *bestG*. The enhancement of accuracy is highest for $O_3$ (up to 22%)
and lowest for $PM_{10}$ (below 3%). We then investigate whether this benefit of ensemble
averaging of air quality time series holds at each station by direct comparison between the

| | |
|---|---|
| 1 | *mme* and the locally best single model (*bestL:* identified from the evaluation at each station). |
| 2 | Summary statistics indicate that the *mme* outscores the *bestL* at roughly 50% of the stations |
| 3 | for $O_3$ and at approximately 40% of the stations for $PM_{10}$, while for $NO_2$ the values were |
| 4 | about 40% and 60% for the two datasets. This result indicates that there is a considerable |
| 5 | amount of stations (over 40%) where the unconditional averaging is not advantageous |
| 6 | because the ensemble does not meet the necessary conditions. A new chart has been |
| 7 | introduced in this paper that interprets the skill of the *mme* according to the skill difference |
| 8 | and the error dependence of the ensemble members. |

| | |
|---|---|
| 9 | The four examined ensemble estimators are then assessed for their skill in the average error as |
| 10 | well as their capability to correctly identify extreme values (events exceeding threshold |
| 11 | value). The key results of the analysis are summarized below: |

| | | |
|---|---|---|
| 12 | - | The skill score of *mme* over its guaranteed upper ceiling (case of zero diversity) ranges |
| 13 | | between 15% and 30%, being lower for $PM_{10}$. Those percentages also represent the |
| 14 | | diversity normalized by the accuracy. Therefore, besides improving the single models, |
| 15 | | their combination in an ensemble confines the *mme* skill if their diversity is limited. |
| 16 | - | The promotion of the right amount of accuracy and diversity in the conditional |
| 17 | | ensemble estimators almost doubles the distance to the guaranteed upper ceiling. The |
| 18 | | skill score over mme is higher for $O_3$ (in the range 18%-31%) and lower for $NO_2$ and |
| 19 | | $PM_{10}$ (in the range 8%-25%), associated to the extent of potential changes in the joint |
| 20 | | distribution of accuracy and diversity in the respective ensembles. The improvement is |
| 21 | | larger for *mmW* and smaller for *mme<* and *kzFO*. |
| 22 | - | The theoretical minimum MSE of *mme* for the case of unbiased and uncorrelated |
| 23 | | models is far from being achieved from all ensemble estimators. |
| 24 | - | As we move towards the tail, the probability of detection (POD) of *bestG* (*bestL*) |
| 25 | | dominates over the *mme (mme<)*. At the extreme percentiles, *kzFO* and *mmW* are the |
| 26 | | only estimators with POD higher than *bestL*. |
| 27 | - | The combination of the results from the average error and the extremes identifies |
| 28 | | *mmW* as the estimator that outscores the others across all percentiles. *kzFO* has high |
| 29 | | capacity for extremes but requires attention for the limited sites with high $N_{EFF}$, where |
| 30 | | its skill is inferior to *mme*. *kzHO* and *mme<* have both high skill across all percentiles |
| 31 | | (better for *kzHO*) but they could have inferior POD compared to *bestL* at the extreme |
| 32 | | percentiles. |

The skill enhancement is superior using the weighting scheme but the required training period to acquire representative weights was longer compared to the sub-selecting schemes. For all pollutants, the variability of the weights and the bias has negligible effect in the error for training periods longer than 60 days. For the schemes relying in member selection, accurate recent representations on the order of a week were sufficient. The learning periods constitute the necessary time to acquire similar prior and posterior distributions in the controlling parameters of samples. The risks of all the statistical learning processes originate from the violation of this assumption, which holds for example in the case of changing weather or chemical regimes. Therefore, the operational implementation of each ensemble approach requires knowledge of its safety margins for the examined pollutants as well as its risks.

The improvement of the physical, chemical and dynamical processes in the deterministic models is a continuous procedure that results in better forecasts. Besides that, mathematical optimizations in the input data (e.g. data assimilation) or the model output (e.g. ensemble estimators) have a significant contribution in the accuracy of the whole modelling process. The presented post-simulation advancements were the result of only favourable ensemble design. However, the theoretical minimum MSE of *mme* for the case of unbiased and uncorrelated models is far from being achieved from all ensemble estimators. Further development is underway in the presented ensemble methods that take into account the meteorological and chemical regimes.

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

**Acknowledgements**
We gratefully acknowledge the contribution of various groups to the second air Quality
Model Evaluation international Initiative (AQMEII) activity: U.S. EPA, Environment
Canada, Mexican Secretariat of the Environment and Natural Resources (Secretaría de Medio
Ambiente y Recursos Naturales-SEMARNAT) and National Institute of Ecology (Instituto
Nacional de Ecología-INE) (North American national emissions inventories); U.S. EPA
(North American emissions processing); TNO (European emissions processing);
ECMWF/MACC project & Météo-France/CNRM-GAME (Chemical boundary conditions).
Ambient North American concentration measurements were extracted from Environment
Canada's National Atmospheric Chemistry Database (NAtChem) PM database and provided
by several U.S. and Canadian agencies (AQS, CAPMoN, CASTNet, IMPROVE, NAPS,
SEARCH and STN networks); North American precipitation-chemistry measurements were
extracted from NAtChem's precipitation-chemistry data base and were provided by several
U.S. and Canadian agencies (CAPMoN, NADP, NBPMN, NSPSN, and REPQ networks); the
WMO World Ozone and Ultraviolet Data Centre (WOUDC) and its data-contributing
agencies provided North American and European ozonesonde profiles; NASA's AErosol
RObotic NETwork (AeroNet) and its data-contributing agencies provided North American
and European AOD measurements; the MOZAIC Data Centre and its contributing airlines
provided North American and European aircraft takeoff and landing vertical profiles; for
European air quality data the following data centers were used: EMEP European Environment
Agency/European Topic Center on Air and Climate Change/AirBase provided European air-
and precipitation-chemistry data. The Finish Meteorological Institute is acknowledged for
providing biomass burning emission data for Europe. Data from meteorological station
monitoring networks were provided by NOAA and Environment Canada (for the US and
Canadian meteorological network data) and the National Center for Atmospheric Research
(NCAR) data support section. Joint Research Center Ispra/Institute for Environment and
Sustainability provided its ENSEMBLE system for model output harmonization and analyses
and evaluation. The co-ordination and support of the European contribution through COST
Action ES1004 EuMetChem is gratefully acknowledged. The views expressed here are those
of the authors and do not necessarily reflect the views and policies of the U.S. Environmental
Protection Agency (EPA) or any other organization participating in the AQMEII project. This
paper has been subjected to EPA review and approved for publication. The UPM authors
thankfully acknowledge the computer resources, technical expertise and assistance provided

by the Centro de Supercomputación y Visualización de Madrid (CESVIMA) and the Spanish Supercomputing Network (BSC). GC and PT were supported by the Italian Space Agency (ASI) in the frame of PRIMES project (contract n. I/017/11/0). The same authors are deeply thankful to the Euro Mediterranean Centre on Climate Change (CMCC) for having made available the computational resources.

**Appendix I**
The relevant separate scales of motion are defined by means of physical considerations and
periodogram analysis (Rao et al., 1997). They are namely the intra-day component (ID), the
diurnal component (DU), the synoptic component (SY) and the long-term component (LT).
The hourly time series (S) can therefore be decomposed as:

$$S(t) = ID(t) + DU(t) + SY(t) + LT(t) \tag{1}$$

where:

$$ID(t) = S(t) - KZ_{3,3}$$

$$DU(t) = KZ_{3,3} - KZ_{13,5}$$

$$SY(t) = KZ_{13,5} - KZ_{103,5} \tag{2}$$

$$LT(t) = KZ_{103,5}$$

**Table 1. The modelling systems participating in the first and second phases of AQMEII for Europe.**

| Model | | | Grid | Emissions | Chemical BC |
|---|---|---|---|---|---|
| | Met | AQ | | | |
| **EU – AQMEII phase I** | MM5 | DEHM | 50 km | Global emission databases, EMEP | Satellite measurements |
| | MM5 | Polyphemus | 24 km | Standard[§] | Standard |
| | MM5 | Chimere | 25 km | MEGAN, Standard | Standard |
| | MM5 | CAMx | 15 km | MEGAN, Standard | Standard |
| | PARLAM-PS | EMEP | 50 km | EMEP model | From ECMWF and forecasts |
| | WRF | CMAQ | 18 km | Standard[§] | Standard |
| | WRF | Chem | 22.5 km | Standard[§] | Fixed |
| | ECMWF | SILAM | 24 km | Standard anthropogenic; In-house biogenic | Standard |
| | ECMWF | Lotos-EUROS | 25 km | Standard[§] | Standard |
| | GEM | GEM-AQ | 25 km | Standard (AQMEII region); EDGAR/GEIA (rest of the global domain) | Global variable grid setup (no boundary conditions) |
| | COSMO | Muscat | 24 km | Standard[§] | Standard |
| | COSMO-CLM | CMAQ | 24 km | Standard[§] | Standard |
| **EU – AQMEII phase II** | WRF | Chem | 23 km | Standard | Standard |
| | WRF | CMAQ | 18 km | Standard | Standard |
| | COSMO | Cosmo-ART | 0.22° | Standard | Standard |
| | COSMO | Muscat | 0.25° | Standard | Standard |
| | NMMB | BSCCTM | 0.20° | Standard | Standard |
| | RACMO | LOTOS-EUROS | 0.5° x 0.25° | Standard | Standard |
| | MetUM | UKCA RAQ | 0.22° | Standard | Standard |

AQMEII phase I
Standard Boundary conditions: provided from GEMS project (Global and regional Earth-system Monitoring using Satellite and in-situ data).
Refer to Schere et al. (2012) for details.
[§] Standard anthropogenic emissions and biogenic emissions derived from meteorology (temperature and solar radiation) and land use
distribution implemented in the meteorological driver. Refer to Solazzo et al. (2012a-b) and references therein for details.
AQMEII phase II
Standard Boundary conditions: 3-D daily chemical boundary conditions were provided by the ECMWF IFS-MOZART model run in the
context of the MACC-II project (Monitoring Atmospheric Composition and Climate - Interim Implementation) at 3-hourly and 1.125 spatial
resolution. Refer to Im et al. (2015a-b) for details.

1 Standard Emissions: based on the TNO-MACC-II (Netherlands Organization for Applied Scientific Research, Monitoring Atmospheric
2 Composition and Climate - Interim Implementation) framework for Europe. Refer to Im et al. (2015a-b) for details.
3

Table 2. The statistical distribution of (a) the Normalized Mean Square Error (NMSE) of the best model (NMSE$_{BEST}$), (b) the ensemble average NMSE (<NMSE>) and (c) the skill difference indicator (NMSE$_{BEST}$ /<NMSE>). In addition, the coefficient of variation (CoV = standard deviation / mean) of the number of cases where each model has been identified as best. All indicators have been evaluated at each monitoring site for the examined species of the two AQMEII phases.

| | O$_3$ (I/II) | O$_3$ (I/II) | NO$_2$ (I/II) | NO$_2$ (I/II) | PM$_{10}$ (I/II) | PM$_{10}$ (I/II) |
|---|---|---|---|---|---|---|
| | <NMSE> | NMSE$_{BEST}$ | <NMSE> | NMSE$_{BEST}$ | <NMSE> | NMSE$_{BEST}$ |
| 5$^{th}$ | 0.04 / 0.04 | 0.03 / 0.03 | 0.28 / 0.23 | 0.17 / 0.18 | 0.30 / 0.27 | 0.20 / 0.20 |
| 25$^{th}$ | 0.07 / 0.07 | 0.05 / 0.05 | 0.39 / 0.35 | 0.24 / 0.25 | 0.40 / 0.39 | 0.26 / 0.28 |
| 50$^{th}$ | 0.10 / 0.10 | 0.07 / 0.08 | 0.52 / 0.49 | 0.33 / 0.34 | 0.47 / 0.51 | 0.34 / 0.37 |
| 75$^{th}$ | 0.15 / 0.15 | 0.11 / 0.12 | 0.82 / 0.76 | 0.48 / 0.50 | 0.61 / 0.62 | 0.46 / 0.50 |
| 95$^{th}$ | 0.24 / 0.23 | 0.18 / 0.18 | 1.69 / 1.49 | 0.81 / 0.93 | 1.02 / 0.98 | 0.73 / 0.81 |
| $\dfrac{NMSE_{BEST}}{<NMSE>}$ | O$_3$ (I) | O$_3$ (II) | NO$_2$ (I) | NO$_2$ (II) | PM$_{10}$ (I) | PM$_{10}$ (II) |
| 5$^{th}$ | 0.50 | 0.60 | 0.36 | 0.45 | 0.49 | 0.63 |
| 25$^{th}$ | 0.62 | 0.70 | 0.50 | 0.62 | 0.61 | 0.72 |
| 50$^{th}$ | 0.70 | 0.76 | 0.61 | 0.72 | 0.70 | 0.79 |
| 75$^{th}$ | 0.76 | 0.82 | 0.72 | 0.81 | 0.85 | 0.85 |
| 95$^{th}$ | 0.83 | 0.88 | 0.87 | 0.93 | 0.92 | 0.92 |
| mean | 0.69 | 0.75 | 0.61 | 0.70 | 0.72 | 0.77 |
| N$_{BEST}$ | O$_3$ (I) | O$_3$ (II) | NO$_2$ (I) | NO$_2$ (II) | PM$_{10}$ (I) | PM$_{10}$ (II) |
| CoV | 1.08 | 0.70 | 1.42 | 0.65 | 1.16 | 1.53 |

Table 3. The MSE from (a) the best deterministic models, globally (*bestG*) and locally (*bestL*), (b) the unconditional ensemble mean (*mme*) and (c) the four conditional ensemble estimators (*mme<, kzFO, kzHO, mmW*). In addition, the bounds for the MSE of the ensemble mean are also presented. The maximum value (<MSE>) arises for ensemble members without diversity and the minimum value (mmeMIN) has been estimated from the variance term only (i.e. calculated for unbiased and uncorrelated ensemble members). The ability of the estimators is evaluated through their skill scores ($SS_{REF}=1-MSE/MSE_{REF}$, REF=bestG, <MSE>, mme).

| O3 (I) | MSE | SS (bestG) | SS (<MSE>) | SS (mme) | O3 (II) | MSE | SS (bestG) | SS (<MSE>) | SS (mme) |
|---|---|---|---|---|---|---|---|---|---|
| bestG | 641 | | 7% | | bestG | 499 | | 14% | |
| bestL | 483 | 25% | 30% | 3% | bestL | 441 | 12% | 24% | 3% |
| mme | 498 | 22% | 28% | | mme | 454 | 9% | 21% | |
| mme< | 398 | 38% | 42% | 20% | mme< | 374 | 25% | 35% | 18% |
| kzFO | 400 | 38% | 42% | 20% | kzFO | 369 | 26% | 36% | 19% |
| kzHO | 367 | 43% | 47% | 26% | kzHO | 349 | 30% | 40% | 23% |
| mmW | 345 | 46% | 50% | 31% | mmW | 315 | 37% | 45% | 31% |
| <MSE> | 690 | | | | <MSE> | 577 | | | |
| mmeMIN | 58 | | | | mmeMIN | 41 | | | |

| NO2 (I) | MSE | SS (bestG) | SS (<MSE>) | SS (mme) | NO2 (II) | MSE | SS (bestG) | SS (<MSE>) | SS (mme) |
|---|---|---|---|---|---|---|---|---|---|
| bestG | 77 | | 25% | | bestG | 61 | | 20% | |
| bestL | 70 | 10% | 32% | 3% | bestL | 58 | 5% | 25% | -4% |
| mme | 72 | 7% | 30% | | mme | 56 | 9% | 27% | |
| mme< | 63 | 19% | 39% | 13% | mme< | 51 | 17% | 34% | 9% |
| kzFO | 62 | 19% | 40% | 13% | kzFO | 52 | 16% | 33% | 8% |
| kzHO | 59 | 24% | 43% | 18% | kzHO | 48 | 21% | 37% | 14% |
| mmW | 56 | 27% | 46% | 22% | mmW | 46 | 25% | 40% | 18% |
| <MSE> | 104 | | | | <MSE> | 77 | | | |
| mmeMIN | 8 | | | | mmeMIN | 6 | | | |

| PM10 (I) | MSE | SS (bestG) | SS (<MSE>) | SS (mme) | PM10 (II) | MSE | SS (bestG) | SS (<MSE>) | SS (mme) |
|---|---|---|---|---|---|---|---|---|---|
| bestG | 341 | | 16% | | bestG | 141 | | 14% | |
| bestL | 326 | 5% | 20% | 1% | bestL | 139 | 2% | 15% | 0% |
| mme | 330 | 3% | 19% | | mme | 139 | 1% | 15% | |
| mme< | 303 | 11% | 25% | 8% | mme< | 121 | 14% | 26% | 13% |

| | | | | | | | | | |
|---|---|---|---|---|---|---|---|---|---|
| kzFO | 299 | 13% | 27% | 10% | kzFO | 122 | 13% | 25% | 12% |
| kzHO | 294 | 14% | 28% | 11% | kzHO | 117 | 17% | 29% | 16% |
| mmW | 284 | 17% | 30% | 14% | mmW | 105 | 26% | 36% | 25% |
| <MSE> | 407 | | | | <MSE> | 164 | | | |
| mmeMIN | 41 | | | | mmeMIN | 12 | | | |

*mme:* unconditional ensemble mean
*mme<:* conditional ensemble mean (Kioutsioukis and Galmarini, 2014)
*kzFO:* conditional spectral ensemble mean with 1st order components (Galmarini et al., 2013)
*kzHO:* conditional spectral ensemble mean with 2nd and higher order components (*kzHO*)
*mmW:* optimal weighted ensemble (Potempski and Galmarini, 2009)

**Table 4. The probability of detection (POD) and false alarm rate (FAR) from (a) the best deterministic models, globally (*bestG*) and locally (*bestL*), (b) the unconditional ensemble mean (*mme*) and (c) the four conditional ensemble estimators (*mme<, kzFO, kzHO, mmW*). Two thresholds were examined for each indicator, corresponding to tail percentiles.**

| O3 (I) | POD | FAR | POD | FAR | O3 (II) | POD | FAR | POD | FAR |
|---|---|---|---|---|---|---|---|---|---|
| threshold | 120 | | 180 | | threshold | 120 | | 180 | |
| bestG | 37.9 | 3.6 | 11.4 | 0.0 | bestG | 19.9 | 1.2 | 1.2 | 0.0 |
| bestL | 54.7 | 3.5 | 19.5 | 0.0 | bestL | 33.2 | 1.5 | 5.4 | 0.0 |
| mme | 39.9 | 2.5 | 12.0 | 0.0 | mme | 22.0 | 1.2 | 0.5 | 0.0 |
| mme< | 53.5 | 2.6 | 18.3 | 0.0 | mme< | 34.9 | 1.3 | 2.4 | 0.0 |
| kzFO | 57.7 | 3.0 | 19.6 | 0.0 | kzFO | 39.1 | 1.5 | 4.4 | 0.0 |
| kzHO | 57.1 | 2.5 | 19.2 | 0.0 | kzHO | 36.9 | 1.2 | 2.3 | 0.0 |
| mmW | 60.6 | 2.6 | 27.2 | 0.0 | mmW | 45.4 | 1.6 | 8.6 | 0.0 |

| NO2 (I) | POD | FAR | POD | FAR | NO2 (II) | POD | FAR | POD | FAR |
|---|---|---|---|---|---|---|---|---|---|
| threshold | 25 | | 50 | | threshold | 25 | | 50 | |
| bestG | 45.9 | 4.6 | 3.8 | 0.2 | bestG | 39.3 | 3.3 | 4.9 | 0.1 |
| bestL | 48.7 | 4.2 | 8.5 | 0.3 | bestL | 41.4 | 3.1 | 8.1 | 0.1 |
| mme | 49.4 | 4.6 | 3.0 | 0.1 | mme | 44.4 | 3.5 | 5.4 | 0.1 |
| mme< | 52.2 | 4.1 | 7.1 | 0.1 | mme< | 47.6 | 3.2 | 7.6 | 0.1 |
| kzFO | 52.7 | 4.1 | 8.4 | 0.1 | kzFO | 46.5 | 3.1 | 9.5 | 0.1 |
| kzHO | 54.2 | 4.0 | 6.8 | 0.1 | kzHO | 49.5 | 3.2 | 9.3 | 0.1 |
| mmW | 57.0 | 4.1 | 14.8 | 0.2 | mmW | 50.9 | 3.1 | 13.5 | 0.1 |

| PM10 (I) | POD | FAR | POD | FAR | PM10 (II) | POD | FAR | POD | FAR |
|---|---|---|---|---|---|---|---|---|---|
| threshold | 50 | | 90 | | threshold | 50 | | 90 | |
| bestG | 25.9 | 2.7 | 1.2 | 0.0 | bestG | 13.0 | 0.4 | 0.0 | 0.0 |
| bestL | 27.8 | 2.3 | 6.9 | 1.2 | bestL | 14.5 | 0.4 | 1.6 | 0.0 |
| mme | 21.6 | 1.8 | 0.4 | 0.0 | mme | 11.4 | 0.4 | 0.0 | 0.0 |
| mme< | 30.6 | 2.3 | 5.6 | 0.1 | mme< | 13.9 | 0.4 | 0.0 | 0.0 |

| | | | | | | | | | |
|---|---|---|---|---|---|---|---|---|---|
| kzFO | 31.1 | 2.3 | 6.9 | 0.1 | kzFO | 14.1 | 0.3 | 0.0 | 0.0 |
| kzHO | 33.2 | 2.4 | 6.1 | 0.1 | kzHO | 13.2 | 0.3 | 0.2 | 0.0 |
| mmW | 35.5 | 2.6 | 13.3 | 0.2 | mmW | 23.9 | 0.4 | 20.8 | 0.0 |

*mme:* unconditional ensemble mean
*mme<:* conditional ensemble mean (Kioutsioukis and Galmarini, 2014)
*kzFO:* conditional spectral ensemble mean with 1st order components (Galmarini et al., 2013)
*kzHO:* conditional spectral ensemble mean with 2nd and higher order components (*kzHO*)
*mmW:* optimal weighted ensemble (Potempski and Galmarini, 2009)

1  **Table 5. The average MSE of *mmW* for various training lengths, calculated for the testing time-**
2  **series (i.e. not-used in the training phase) that contains all stations.**

| Length of training period (days) | $O_3$ (I) | $O_3$ (II) | $NO_2$ (I) | $NO_2$ (II) | $PM_{10}$ (I) | $PM_{10}$ (II) |
|---|---|---|---|---|---|---|
| 5 | 616 | 540 | 90 | 91 | 717 | 210 |
| 10 | 496 | 441 | 77 | 66 | 443 | 150 |
| 20 | 400 | 378 | 65 | 56 | 348 | 125 |
| 30 | 380 | 344 | 62 | 52 | 308 | 109 |
| 40 | 366 | 334 | 59 | 50 | 300 | 113 |
| 50 | 357 | 326 | 57 | 48 | 294 | 108 |
| 60 | 351 | 319 | 56 | 45 | 282 | 102 |

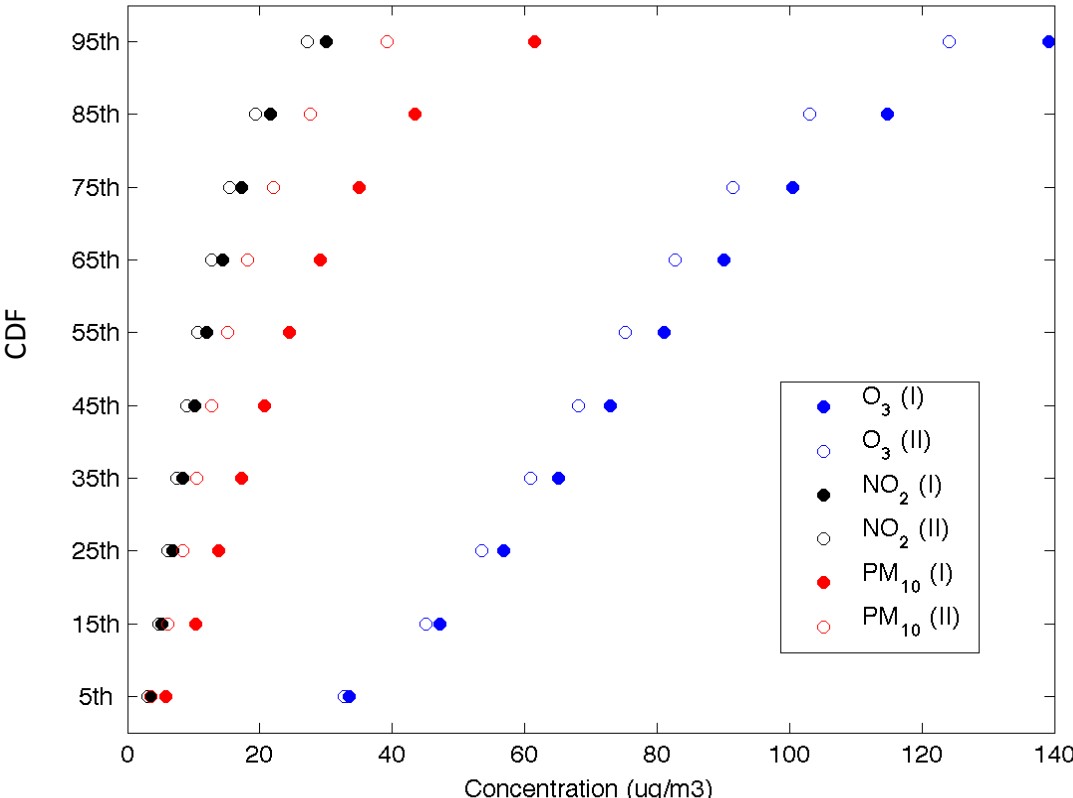

Figure 1. The Cumulative density functions of the observations ($O_3$, $NO_2$, $PM_{10}$) in the two AQMEII phases (Phase I: *filled circles*, Phase II: *non-filled circles*). Each bullet represents the median at the specific percentile.

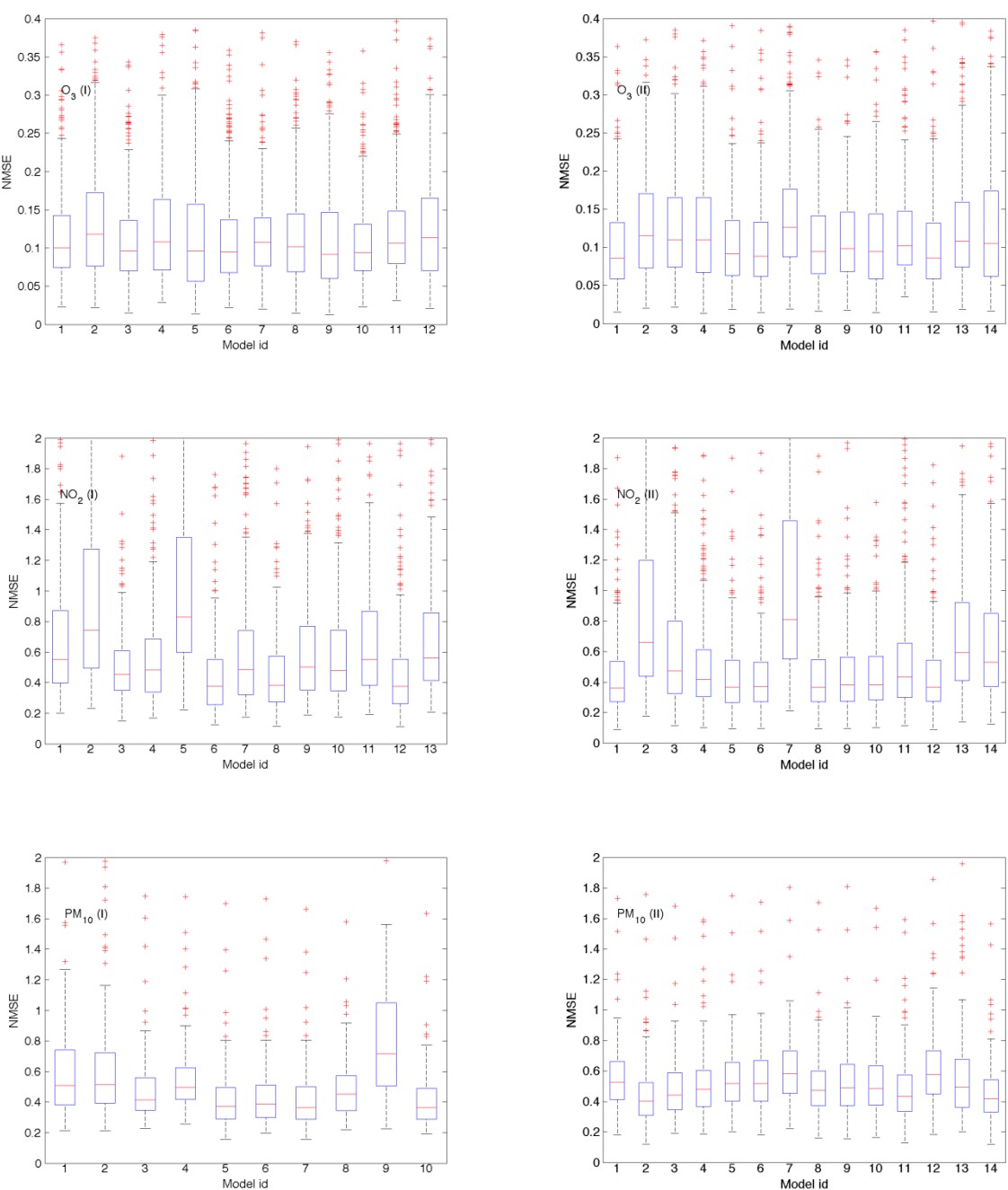

**Figure 2. Model skill difference via the NMSE. On each box, the central mark indicates the median,**
**and the bottom and top edges of the box indicate the 25th and 75th percentiles, respectively. The**
**whiskers extend to the most extreme data points not considered outliers and the outliers (points**
**with distance from the 25th and 75th percentiles larger than 1.5 times the interquartile range) are**
**plotted individually using the '+' symbol.**

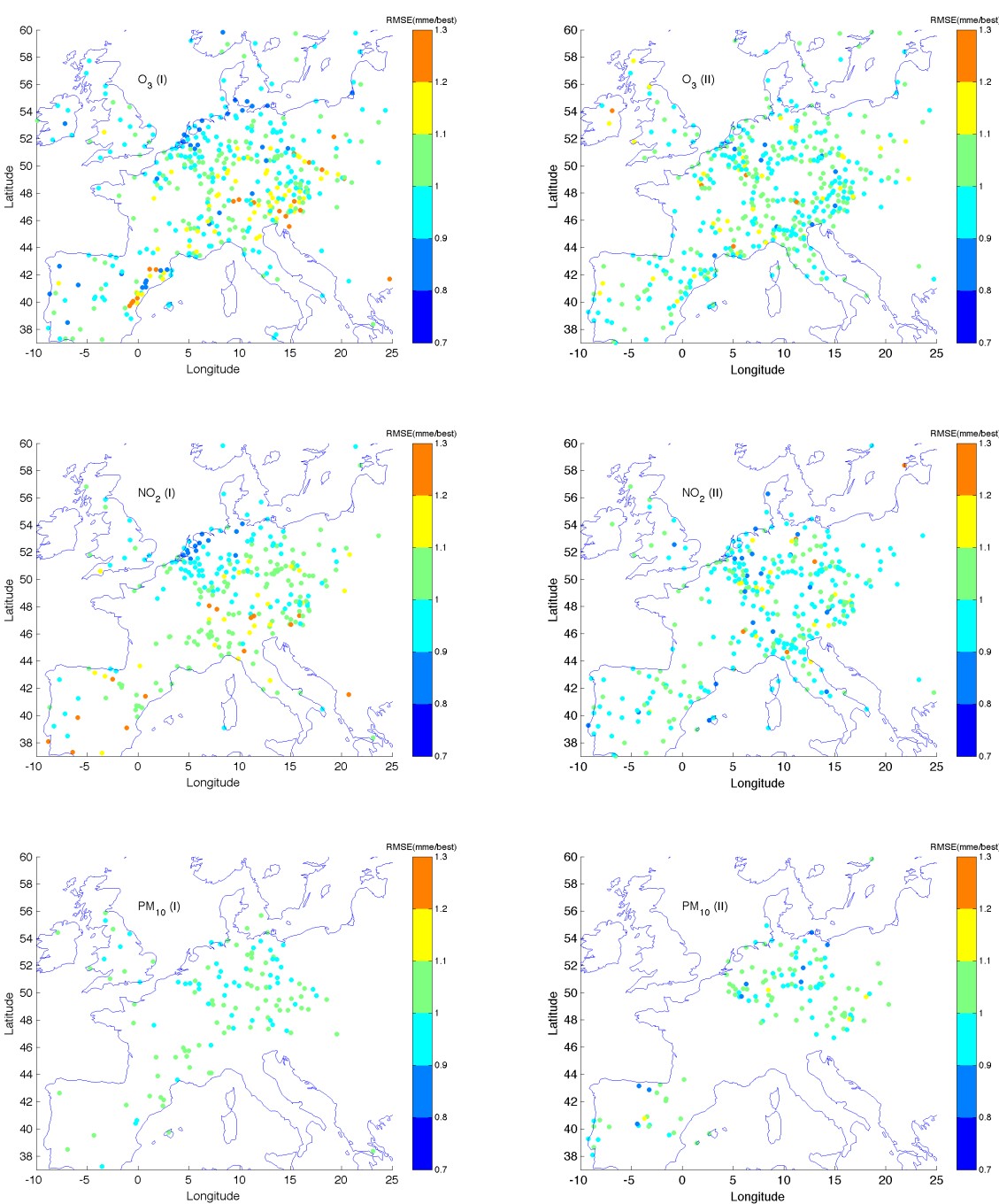

Figure 3. Comparison of the *mme* skill against the best local deterministic model by means of the indicator $RMSE_{MME}/RMSE_{BEST}$.

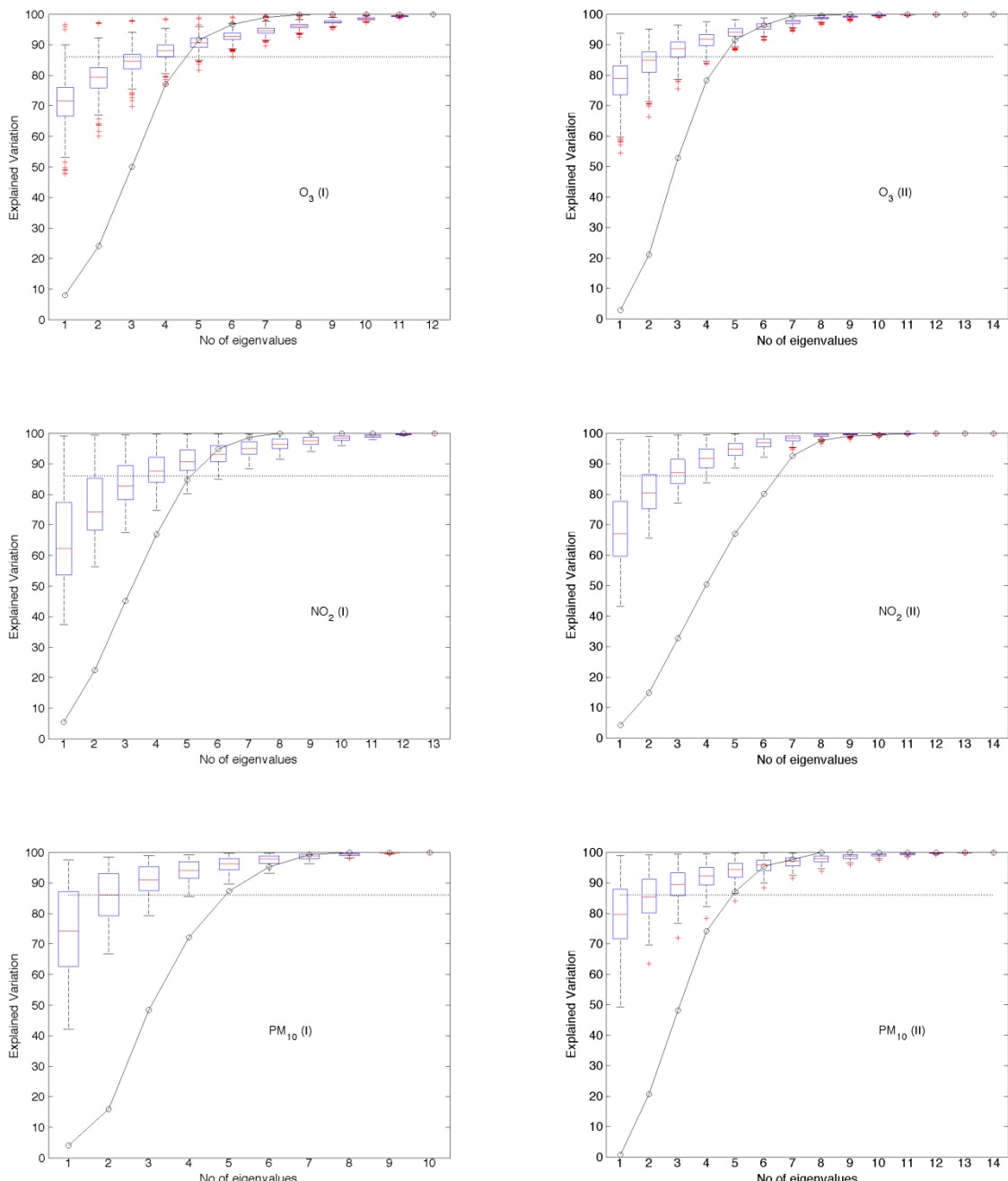

**Figure 4. Model error dependence through the eigenvalues spectrum. The average explained**
**variation from the maximum eigenvalue is 71/78 (phase I/II) for O$_3$, 65/69 for NO$_2$ and 74/79 for**
**PM$_{10}$. On the same graph, the cumulative density function of N$_{EFF}$ calculated from all possible**
**ensemble combinations is presented with the black line.**

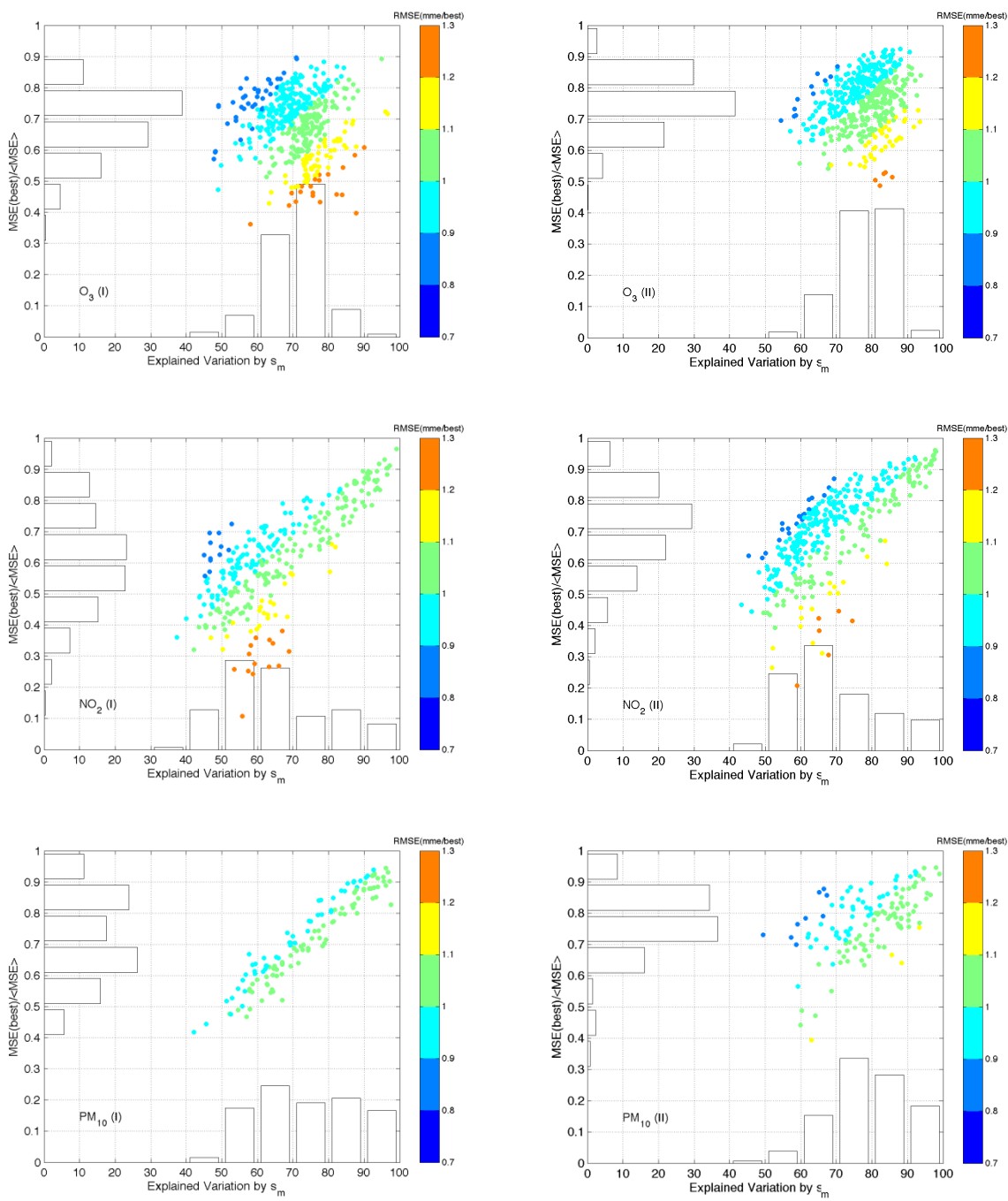

**Figure 5. Interpretation of Figure 4: the explanation of the mme skill against the best local deterministic model with respect to skill difference (evaluated from $MSE_{BEST}/<MSE>$) and error dependence (evaluated from the explained variation by the highest eigenvalue).**

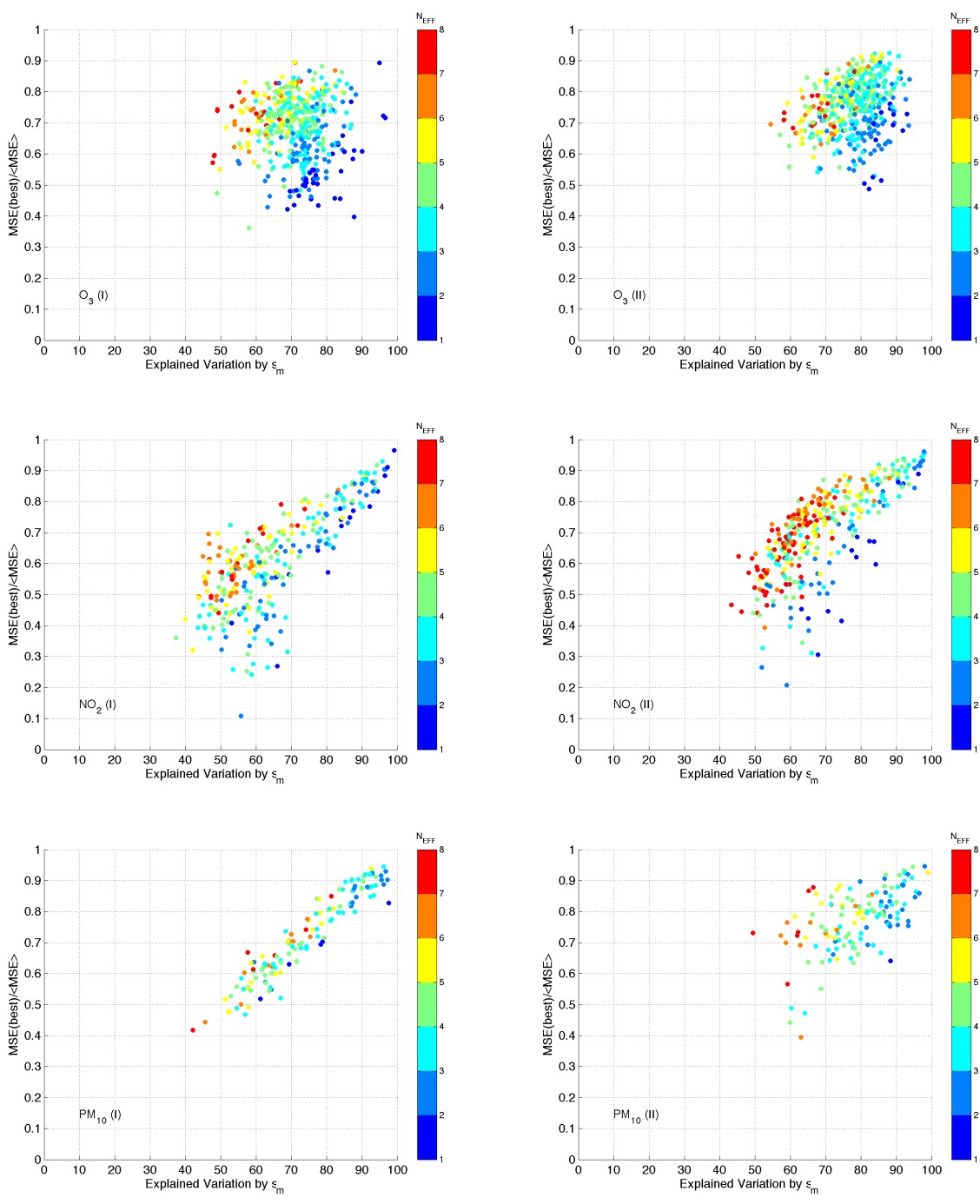

1 **Figure 6. Like Figure 5 but showing the $N_{EFF}$ with respect to skill difference and error dependence.**

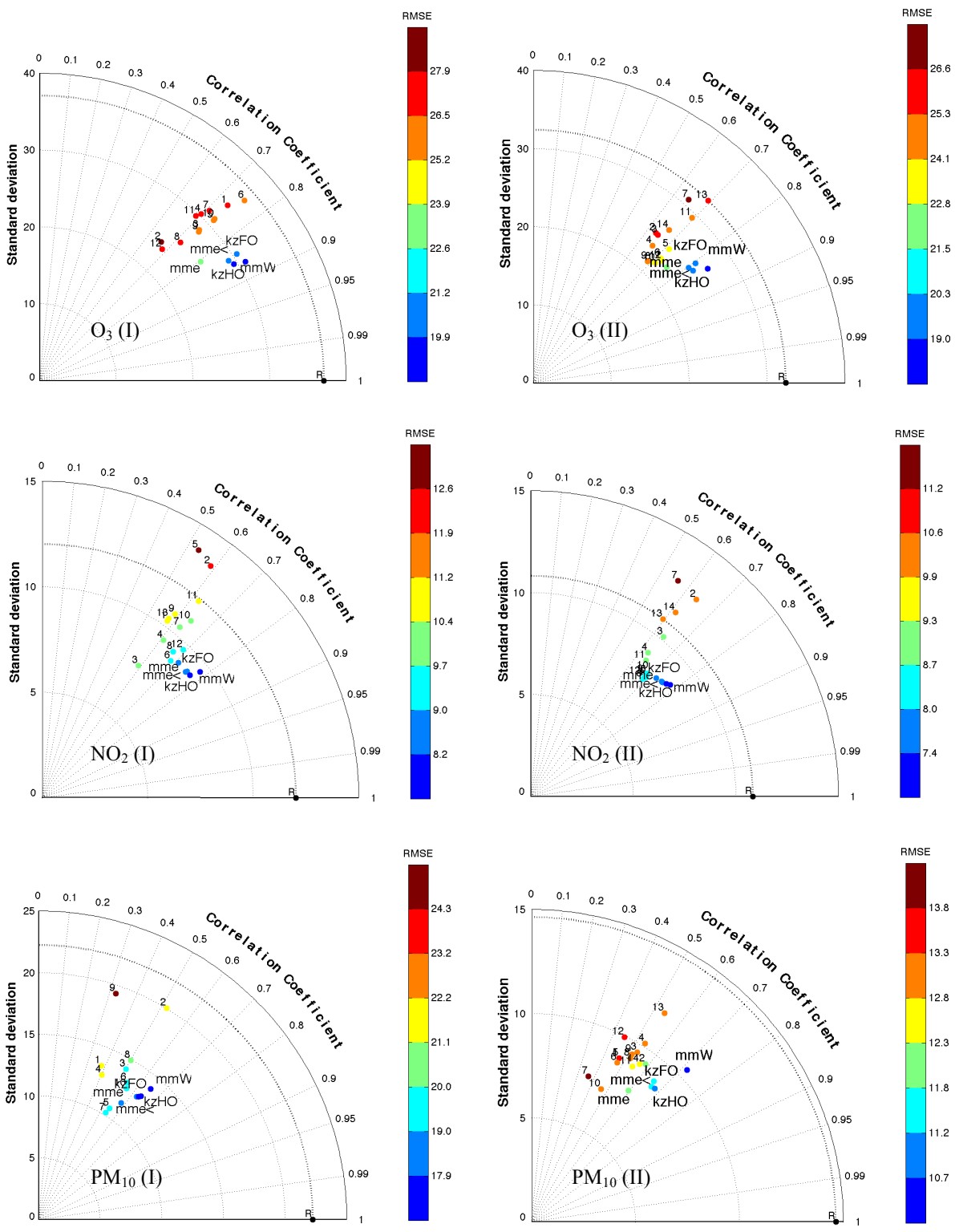

**Figure 7. Composite skill of all deterministic models and ensemble estimators (*mme, mme<, kzFO, kzHO, mmW*) through Taylor plots. The point R represents the reference point (i.e. observations).**

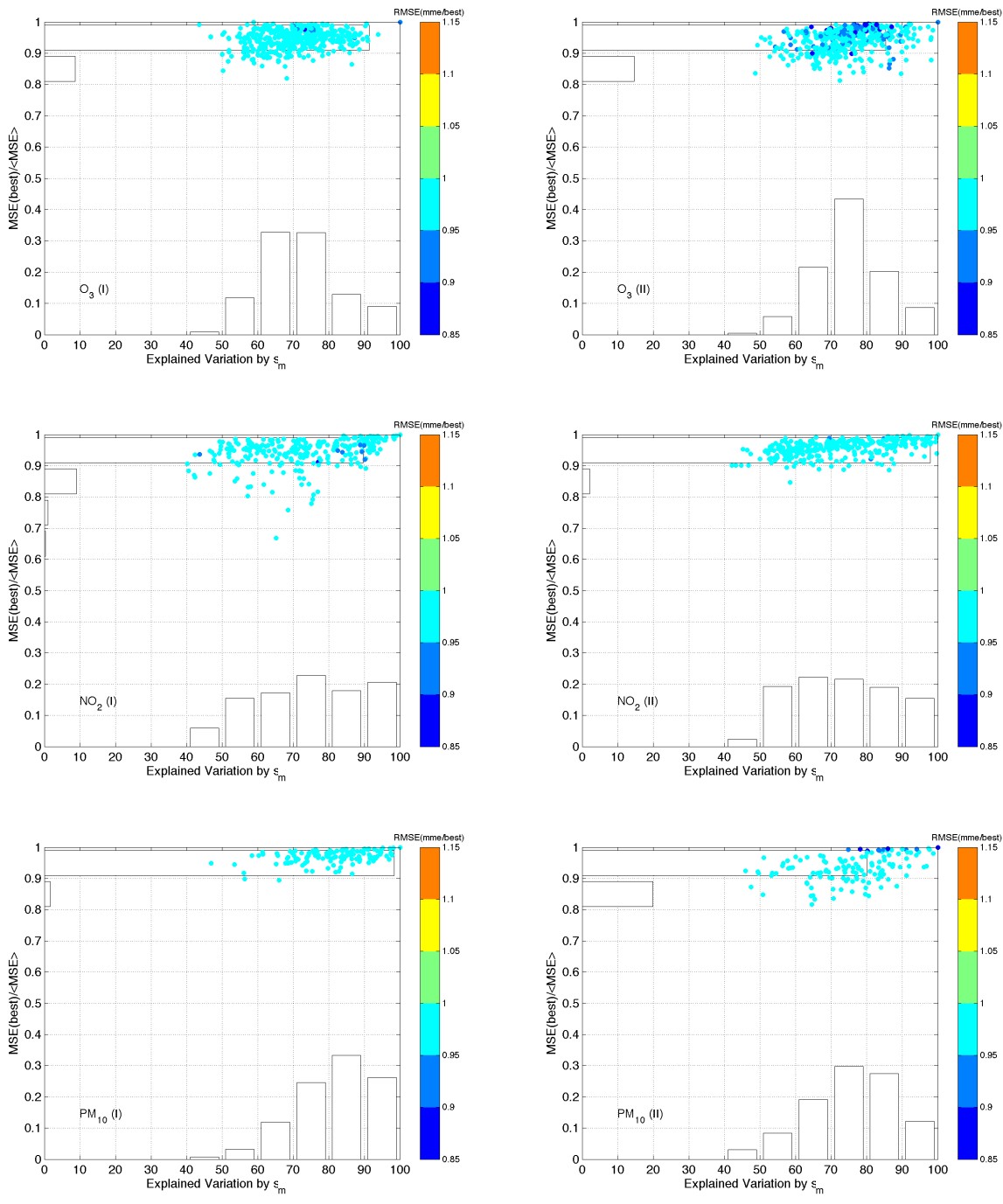

Figure 8. Like Figure 5 but for the *mme<* skill in the reduced ensemble. Please note the change in the colorscale.

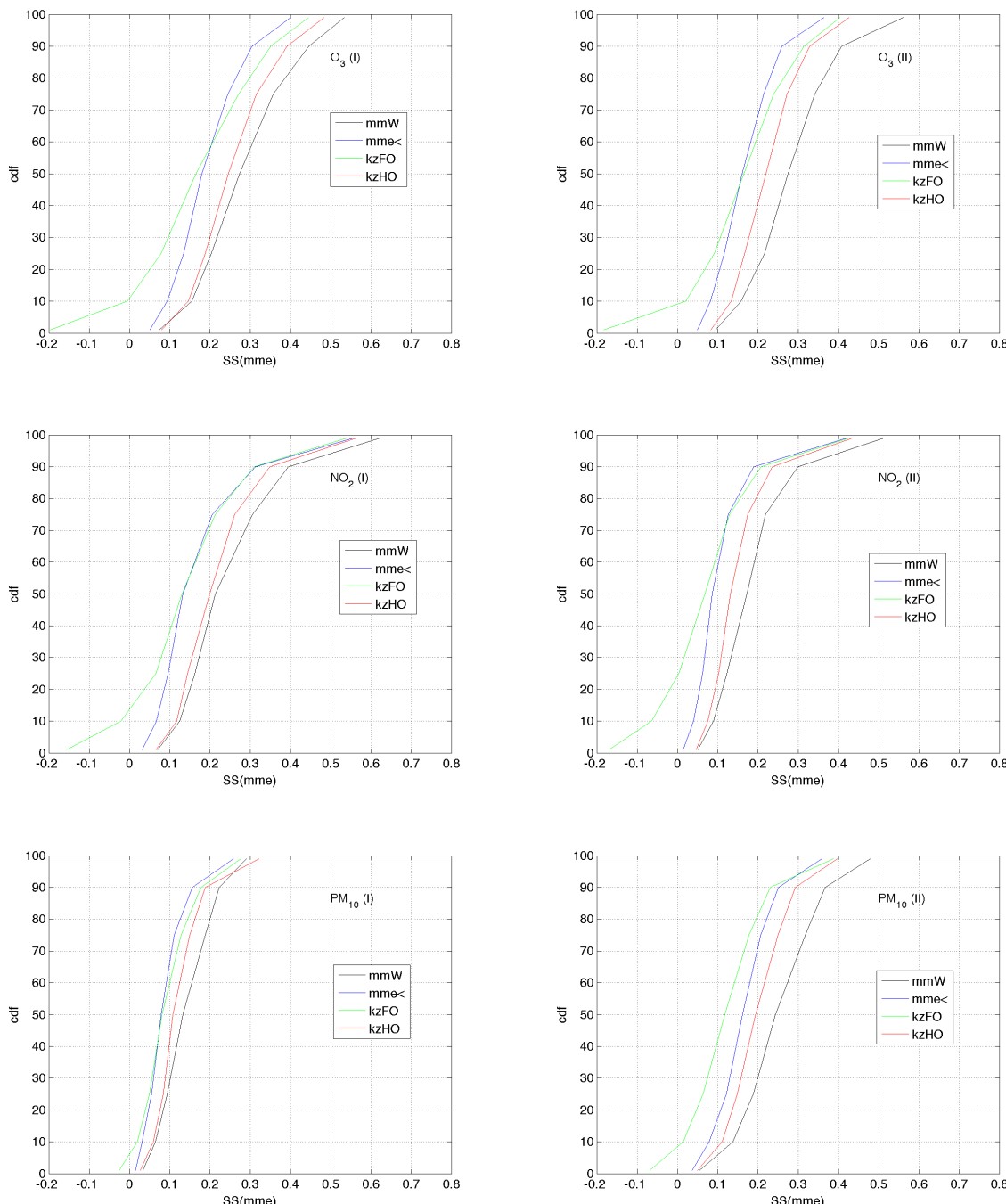

**Figure 9. The cumulative density function of the Skill Score (1-MSE$_X$/MSE$_{MME}$, X = *mmW, mme<,*** *kzFO, kzHO*) over *mme*, evaluated at each monitoring site for the examined species of the two** **AQMEII phases.**

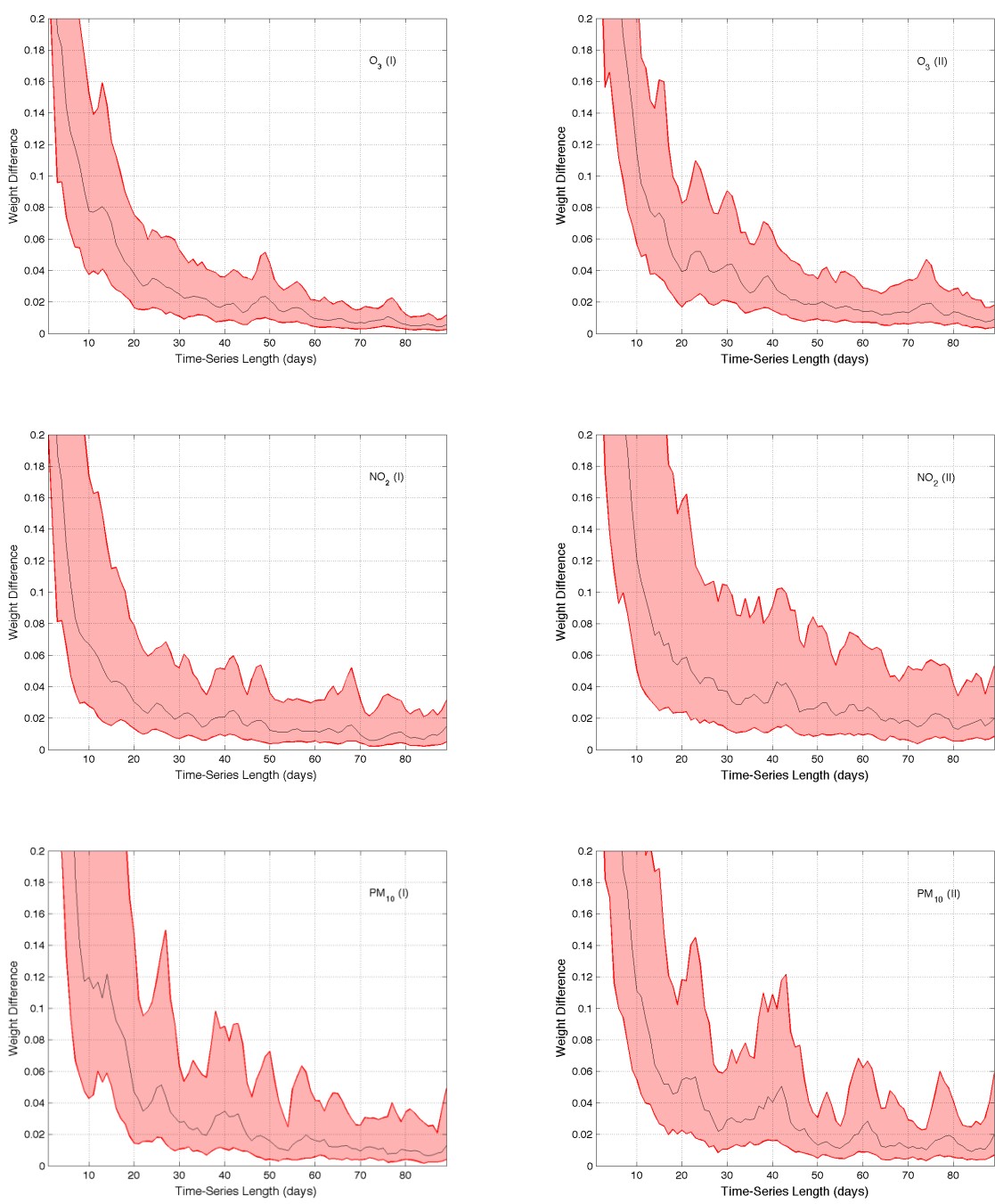

1   **Figure 10. The interquartile range over all stations of the day-to-day difference in the weights**
2   **arising from variable time-series length.**

