# Peer review of "Insights into the deterministic skill of air quality ensembles from the"

_Atmospheric Chemistry and Physics, 2016_

## Referee Comment (RC1) · M. Plu (Referee) · 18 Jul 2016

General comments

The manuscript entitled "Improving the deterministic skill of air quality ensembles" reports about the properties and the scores of different ensembles applied on the model outputs of the two phases of AQMEII (2006 and 2010) over Europe. The presentation of the manuscript and of figures is good. To my view, the study has several merits and the results that are presented are original enough to be published. However, the manuscript should be improved along the following recommendations in order to go over the Discussion step.

[Figure]

Specific comments

1. There is a lack of focus of the manuscript on the main relevant original ideas that are demonstrated. Many interesting results are presented, and the manuscript needs to focussed on one or two main new scientific questions that the manuscript addresses. These lines should be followed from the abstract to the conclusion. To make my argument more understandable, I would like to point out the following:

- the results that are reported in the first paragraph of the abstract - from lines 6 to 10 - are not new as they were demonstrated in past research articles. These lines mislead the reader on the purpose of the article;

- many times (page2-line12, page4-line4, page7-line12), "two ensembles" appear in the text, but it is not clear whether it refers to two different ensemble methods (and actually, the article compares more than two methods) or to the two AQMEII phases.

A suggestion would be to present the article as a comparison of different ensemble methods (mmW, mmS, KZ, ...), applied on two different datasets (both phases of AQMEII). Actually, the manuscript does not present new methods, but it compares the performance of existing ensemble methods on different pollutants and on different periods. What I consider also to be original are the diagnostics (such as Figure 5) that have been developed and that are used to analyse the ensemble properties. The objectives written page 4 (lines 20-23) may also be the relevant lines to follow, which is not fully obvious in the present manuscript.

2. In the same line of thoughts, the title of the manuscript is too general and it should be more specific. A general title such as the one that appears now could apply to many papers that have already been published.

3. Comparing the ensemble performance between the two AQMEII phases does not bring much to the study and can be misleading, since:

- the observation dataset changes (no PM10 observations available from UK nor France in Phase II, page8-lines7-8),

- the period (meteorological regimes, types of pollution, etc) change,

- the individual models change in depth.

The differences in ensemble performances between phases I and II (page9 for instance) are subject to all these differences. The attribution of differences of ensemble performances between the two phases should be done cautiously, making only one variable change at each time for any interpretation. If it is not possible, I suggest then to remove the discussions about the differences between Phase I and Phase II of AQMEII.

4. There is a lack of description of the experimental setup of the two phases of AQMEII, that would help the reader to understand some of the conclusions that are drawn, such as the arguments at page9-line19, page16-line, among others. The manuscript as it is written now is not self-consistent. To improve this, I would suggest to add in section 3.2 the key facts of both AQMEII phases: general experimental setup (domain, periods, common input data and setup for all models) and the different models that participate (name, chemical and aerosol schemes, resolution, meteorological model, etc). At least the key facts that are needed to understand the discussions should appear in the manuscript. For the rest, the manuscript should cite some AQMEII reference articles.

Minor comments

- page 3, if the "Recent results" (line 7) refer to the citation (Eskes, 2002) (line 9), then the word "Recent" does not apply; if they refer to an actual recent other work, please cite it,

- the manuscript would gain in clarity if the KZ methods (page6) were described more in depth; for instance the page12-lines(7-10) sentence is somehow enigmatic.

**[ACPD](ACPD)**
- Is the quarter (September-October-November) chosen for NO2 the most relevant one? Do not the December-January-February quarters show higher NO2 concentration levels?

- The sentence page7-line22 would better fit in section 3.1.

- page8-lines5-8: the sentence "the decline ... due to .. sampling stations." should be proven by some diagnoses or adequate citation.

- page10-line 2: the sentence "the benefits of ensemble ... members)." is not fully clear and maybe not true: what happens if we take the 6 "worse performing" models?

- page10-line17: reference to the relevant figure is needed.

- page17-line28: remove '*'

---

## Referee Comment (RC2) · Anonymous Referee #2 · 23 Aug 2016

The paper analyses the two phases of the AQMEII initiatives to test different techniques for improving deterministic estimates from multi-model ensembles. Even though the paper is generally well written, my opinion is that the scientific novelty is scarce, and most of the conclusions are not solid. Here are my motivations:

1) As stated in the Abstract: Line 5-7 "we demonstrate. . .is far from optimum,". This has been already proved several times is previous publications. (see Solazzo et al. 2013, Riccio et al. 2007, Galmarini et al. 2013, among others). In these papers, the same concepts and techniques of reducing the dimensionality of multi-model ensembles and optimal combination have been widely and repeatedly presented.

2) Pag 4 line 4-13. The differences between the two experiments are described. The differences in the meteorology (two different years) and stations (amount of observa-

tions and their locations) are those that undermine more the statistical significance of the results. Most of them are presented (see Table2 and Table 4) without bootstrap confidence intervals or other techniques to assess if the differences between the two phases are statistically significant. The numbers of Phase I and II are often very close, and despite that, the authors build many conclusions on the top of these small differences. Also, most of the differences (if any) could be explained by the meteorology or the underlying changes in the station network. The authors should, at least, have made an attempt to make the two experiments more homogenous, i.e. by keeping a similar kind of stations over the two phases (same amount of urban, background stations).

3) Section 4.1 Forecasting performances. The authors want to prove that the weighting scheme might be used in forecasting mode. There are two issues here that undermine the conclusions of this section. My understanding is that some of the models participating at the inter-comparison are not running in forecasting mode (they use meteorological reanalysis as boundary conditions). While they should run as an operational real-time forecasting model to be considered as realistic forecasts. Running these model in forecasting mode would change the model behaviors and error structures. Hence the conclusions achieved might change as well. How the bias of the models is removed in this test? Using the bias computed over the entire period (as previously mentioned) to correct forecast issued over the same test period would not be possible in real-time forecasting. This simple bias removal technique might not be so effective especially in forecasting mode when data from the future cannot be used.

Minor comments:

The sentence: "In addition, mathematical tools such as ensemble forecasting provide an extra channel for uncertainty quantification and eventually reduction. Such method seems similar to the Monte Carlo approach; in practice, the similarity is only phenomenological since the probability density function of the uncertainty is not sampled in any statistical context like random, latin-hypercube, etc." is not clear at all. Ensemble forecasting cannot be considered as a mathematical tool in general. What does it

mean:" Similarity in only phenomenological. . .."?

"benefits from ensemble forecasting arise from the averaging out of the unpredictable components (Kalnay, 2003)." It would be correct to say that benefits arise from averaging estimates with uncorrelated errors.

Pag 3 line 25 "One of the challenges in ensemble forecasting is the processing of the deterministic models". This is true only if you are talking about a multi-model ensemble.

Eq1 bias, var, cov? Should be presented with a more detailed notation

Eq 2 E is the mean over what?

Line 6 page 8 keep the same stations over the two phases

Line 24 page 8 indirect feedback of what? Some details should be added

Line 19 page 5 I'd say the minimum (what does it mean ideal?)

Section 2.1 86 % is a general value or something related to this paper

The same bunch of authors (or most of them) appears in previous publications regarding AQMEII phase I and II. I have some doubts (but I might be wrong) that they all give an active contribution to this paper or at least original compared to what already provided in the previous publications regarding these experiments. It would be fair to include in detail a description of the contribution of each author to this paper.

---

## Author Comment (AC1) · 6 Oct 2016

*Referee #1:* **Matthieu. Plu**

We thank Dr Matthieu Plu for the positive and helpful comments that have improved the manuscript. They have all been taken on board and addressed in the revised version of our manuscript.

*General Comments:*

The manuscript entitled "Improving the deterministic skill of air quality ensembles" reports about the properties and the scores of different ensembles applied on the model outputs of the two phases of AQMEII (2006 and 2010) over Europe. The presentation of the manuscript and of figures is good. To my view, the study has several merits and the results that are presented are original enough to be published. However, the manuscript should be improved along the following recommendations in order to go over the Discussion step.

*Specific Comments:*

1. There is a lack of focus of the manuscript on the main relevant original ideas that are demonstrated. Many interesting results are presented, and the manuscript needs to focussed on one or two main new scientific questions that the manuscript addresses. These lines should be followed from the abstract to the conclusion. To make my argument more understandable, I would like to point out the following:

- the results that are reported in the first paragraph of the abstract - from lines 6 to 10 - are not new as they were demonstrated in past research articles. These lines mislead the reader on the purpose of the article;

- many times (page2-line12, page4-line4, page7-line12), "two ensembles" appear in the text, but it is not clear whether it refers to two different ensemble methods (and actually, the article compares more than two methods) or to the two AQMEII phases.

A suggestion would be to present the article as a comparison of different ensemble methods (mmW, mmS, KZ, ...), applied on two different datasets (both phases of AQMEII). Actually, the manuscript does not present new methods, but it compares the performance of existing ensemble methods on different pollutants and on different periods. What I consider also to be original are the diagnostics (such as Figure 5) that have been developed and that are used to analyse the ensemble properties. The objectives written page 4 (lines 20-23) may also be the relevant lines to follow, which is not fully obvious in the present manuscript.

Response: Thank you for the valuable suggestion. The manuscript has been rewritten as an analysis of the performance of different ensemble techniques rather than a comparison of the results from the two phases of the AQMEII activity, focusing on the originality of the study that includes: (a) the comparison of several ensemble methods on pollutants of different skill using different datasets, (b) the introduction of an approach based on high-dimension spectral optimization, (c) the introduction of innovative charts for the interpretation of the error of the unconditional ensemble mean with respect to indicators reflecting the skill difference and error dependence of the models as well as the effective number of models.

2. In the same line of thoughts, the title of the manuscript is too general and it should be more specific. A general title such as the one that appears now could apply to many papers that have already been published.

Response: Done as suggested. The title has been changed to "Insights in the deterministic skill of air quality ensembles from the analysis of AQMEII data"

3. Comparing the ensemble performance between the two AQMEII phases does not bring much to the study and can be misleading, since:

- the observation dataset changes (no PM10 observations available from UK nor France in Phase II, page8-lines7-8),

- the period (meteorological regimes, types of pollution, etc) change,

- the individual models change in depth.

The differences in ensemble performances between phases I and II (page9 for instance) are subject to all these differences. The attribution of differences of ensemble performances between the two phases should be done cautiously, making only one variable change at each time for any interpretation. If it is not possible, I suggest then to remove the discussions about the differences between Phase I and Phase II of AQMEII.

Response: Done. The new presentation is after an analysis of the performance of different ensemble techniques rather than a comparison of the results from the two phases of the AQMEII activity.

4. There is a lack of description of the experimental setup of the two phases of AQMEII, that would help the reader to understand some of the conclusions that are drawn, such as the arguments at page9-line19, page16-line, among others. The manuscript as it is written now is not self-consistent. To improve this, I would suggest to add in section 3.2 the key facts of both AQMEII phases: general experimental setup (domain, periods, common input data and setup for all models) and the different models that partici- pate (name, chemical and aerosol schemes, resolution, meteorological model, etc). At least the key facts that are needed to understand the discussions should appear in the manuscript. For the rest, the manuscript should cite some AQMEII reference articles.

Response: Done as suggested. A new section 3 with the experimental setup has been added.

*Minor comments:*

- page 3, if the "Recent results" (line 7) refer to the citation (Eskes, 2002) (line 9), then the word "Recent" does not apply; if they refer to an actual recent other work, please cite it,

Response: Done as suggested. The word 'recent' has been removed.

- the manuscript would gain in clarity if the KZ methods (page6) were described more in depth; for instance the page12-lines(7-10) sentence is somehow enigmatic.

Response: Done as suggested. Two paragraphs have been introduced in section 2. The first describes the rationale behind spectral optimization (spectral decomposition equations are provided in the Appendix). The second presents the examined ensemble estimators with reference to their theoretical basis.

- Is the quarter (September-October-November) chosen for NO2 the most relevant one? Do not the December-January-February quarters show higher NO2 concentration levels?

Response: We choose a continuous seasonal time series for each pollutant.

- The sentence page7-line22 would better fit in section 3.1.

Response: Done as suggested.

- page8-lines5-8: the sentence "the decline ... due to .. sampling stations." should be proven by some diagnoses or adequate citation.

Response: The sentence has been removed from the text.

- page10-line 2: the sentence "the benefits of ensemble ... members)." is not fully clear and maybe not true: what happens if we take the 6 "worse performing" models?

Response: We have rephrased the sentence to emphasize that we do not mean particular models.

- page10-line17: reference to the relevant figure is needed.

Response: The sentence has been removed from the text.

- page17-line28: remove '*'

Response: Done as suggested.

---

## Author Comment (AC2) · 6 Oct 2016

*Referee #2: Anonymous*

We thank the anonymous reviewer for the many helpful suggestions that have improved the manuscript. They have all been taken into consideration and addressed in the revised version of our manuscript.

*General/Specific Comments:*

The paper analyses the two phases of the AQMEII initiatives to test different techniques for improving deterministic estimates from multi-model ensembles. Even though the paper is generally well written, my opinion is that the scientific novelty is scarce, and most of the conclusions are not solid. Here are my motivations:

1) As stated in the Abstract: Line 5-7 "we demonstrate. . .is far from optimum,". This has been already proved several times is previous publications. (see Solazzo et al. 2013, Riccio et al. 2007, Galmarini et al. 2013, among others). In these papers, the same concepts and techniques of reducing the dimensionality of multi-model ensembles and optimal combination have been widely and repeatedly presented.

Response: We have re-written many parts of the manuscript to make more clear the focus and originality of the study. The scientific novelty of the study includes: (a) the comparison of several ensemble methods on pollutants of different skill using different datasets, (b) the introduction of an approach based on high-dimension spectral optimization, (c) the introduction of innovative charts for the interpretation of the error of the unconditional ensemble mean with respect to indicators reflecting the skill difference and error dependence of the models as well as the effective number of models. The manuscript has been rewritten to better reflect its originality.

2) Pag 4 line 4-13. The differences between the two experiments are described. The differences in the meteorology (two different years) and stations (amount of observations and their locations) are those that undermine more the statistical significance of the results. Most of them are presented (see Table2 and Table 4) without bootstrap confidence intervals or other techniques to assess if the differences between the two phases are statistically significant. The numbers of Phase I and II are often very close, and despite that, the authors build many conclusions on the top of these small differences. Also, most of the differences (if any) could be explained by the meteorology or the underlying changes in the station network. The authors should, at least, have made an attempt to make the two experiments more homogenous, i.e. by keeping a similar kind of stations over the two phases (same amount of urban, background stations).

Response: We have tested the statistical hypotheses on the differences of the distributions and their means through the Kolmogorov-Smirnov test and the t-test respectively. Although many differences were generally significant at the 1% level, we have decided to remove the comparisons of the two phases. The two experiments were independently designed and executed and have many differences. The harmonization of the validation set would remove an uncertain factor. Even then, the attribution of the differences between the two datasets to the uncertain factors (meteorology, models, coupling, etc.) in a statistical framework would still include a considerable amount of uncertainty. Moreover, such quantitative decomposition is beyond the objectives stated in this study. Therefore, the manuscript has been rewritten as an analysis of the performance of different ensemble techniques rather than as a comparison of the results from the two phases of the AQMEII activity.

3) Section 4.1 Forecasting performances. The authors want to prove that the weighting scheme might be used in forecasting mode. There are two issues here that undermine the conclusions of this section. My understanding is that some of the models participating at the inter-comparison are not running in forecasting mode (they use meteorological reanalysis as boundary conditions). While they

should run as an operational real-time forecasting model to be considered as realistic forecasts. Running these model in forecasting mode would change the model behaviors and error structures. Hence the conclusions achieved might change as well. How the bias of the models is removed in this test? Using the bias computed over the entire period (as previously mentioned) to correct forecast issued over the same test period would not be possible in real-time forecasting. This simple bias removal technique might not be so effective especially in forecasting mode when data from the future cannot be used.

Response: With respect to the first issue, the term 'forecast' has been changed to 'simulation' throughout the text. Concerning the second issue, bias removal is beneficial to the ensemble mean according to the bias-variance-covariance decomposition. It is not necessary for the approaches relying on reduced-dimensionality ensembles but the formulas for the analytically optimized weights have been derived with the assumption of bias-free members. As for the implementation, the mean bias over the training period is removed from the time-series of the test dataset. An explanation has been added in the text to clarify that the bias calculated in the train dataset (for the examined training periods of 5-60 days) is subtracted from the test dataset.

Our results indicate that after 30-60 days, the variable biases and weights have no effect in the skill of the weighted ensemble mean. Besides that, the seasonal bias reflects the systematic errors of the single models and it is considered a known quantity for validated models. Those considerations support the possible application of the approaches in real-time forecasting.

*Minor comments:*

The sentence: "In addition, mathematical tools such as ensemble forecasting provide an extra channel for uncertainty quantification and eventually reduction. Such method seems similar to the Monte Carlo approach; in practice, the similarity is only phenomenological since the probability density function of the uncertainty is not sampled in any statistical context like random, latin-hypercube, etc." is not clear at all. Ensemble forecasting cannot be considered as a mathematical tool in general. What does it mean:" Similarity in only phenomenological. . ..."?

Response: The sentence has been removed from the text.

"benefits from ensemble forecasting arise from the averaging out of the unpredictable components (Kalnay, 2003)." It would be correct to say that benefits arise from averaging estimates with uncorrelated errors.

Response: Done. The sentence has changed accordingly.

Pag 3 line 25 "One of the challenges in ensemble forecasting is the processing of the deterministic models". This is true only if you are talking about a multi-model ensemble.

Response: Done. The sentence has changed accordingly.

Eq1 bias, var, cov? Should be presented with a more detailed notation

Response: Done.

Eq 2 E is the mean over what?

Response: Eq 1 and Eq 2 are related through their expectations over multiple stations.

Line 6 page 8 keep the same stations over the two phases

Response: We do not compare differences between the two phases in the revised manuscript.

Line 24 page 8 indirect feedback of what? Some details should be added

Response: The sentence has been removed from the text.

Line 19 page 5 I'd say the minimum (what does it mean ideal?)

Response: Done. The sentence has changed accordingly.

Section 2.1 86 % is a general value or something related to this paper

Response: It is general, the first $N_{EFF}$ members account for 86% of the variability.

The same bunch of authors (or most of them) appears in previous publications regarding AQMEII phase I and II. I have some doubts (but I might be wrong) that they all give an active contribution to this paper or at least original compared to what already provided in the previous publications regarding these experiments. It would be fair to include in detail a description of the contribution of each author to this paper.

Response: The authors present in many AQMEII publications, as well as this one, are from the modeling groups that performed the simulations. Without the simulations, none of the published analyses would be possible.